# Assimilation of satellite NO$_2$ observations at high spatial resolution using OSSEs

X. Liu[1], A. Mizzi[2], J. Anderson[3], I. Fung[1] and R.C. Cohen[1,4]

[1]Department of Earth and Planetary Science, University of California at Berkeley, Berkeley, CA, USA
[2]Atmospheric Chemistry Observation and Modeling Laboratory, National Center for Atmospheric Research, Boulder, CO, USA
[3]Institute for Mathematics Applied to Geosciences, National Center for Atmospheric Research, Boulder, CO, USA
[4]Department of Chemistry, University of California at Berkeley, Berkeley, CA, USA

*Correspondence to*: Ronald C. Cohen (rccohen@berkeley.edu)

**Abstract.** Observations of trace gases from space-based instruments offer the opportunity to constrain chemical and weather forecast and reanalysis models using the tools of data assimilation. To date attempts at assimilation of nitrogen dioxide (NO$_2$) satellite remote sensing data have focused on updating emissions and concentrations. These initial efforts evaluated updates at length scales of ~100 km using once a day measurements from satellites with ground pixels of 13 km × 24 km or
larger. In the boundary layer, NO$_2$ has a lifetime of order five hours and corresponding e-folding concentration variations near urban and point sources occur on spatial scales of order 50-75 km. Accurate observation and modeling of these variations require spatial resolution of order 4 km. In addition because of the short lifetime, NO$_2$ variations are more strongly coupled to short time scale meteorological parameters than longer lived chemicals such as CO or CO$_2$. In the next few years, we anticipate the launch of several instruments with ~3 km spatial resolution. In addition, some of these instruments will be
in geostationary orbits and thus have hourly revisit times. In anticipation of these instruments, we investigate the potential of high space and time resolution column measurements to serve as constraints on urban NO$_x$ emissions using a geostationary observation simulator coupled to a data assimilation system. Our results show the dependence of errors in estimated emissions on the wind forecast errors. Modeled wind vectors must have RMSE below 1 m/s for the success of chemical assimilation to infer NO$_x$ emissions at the scale of the model grid with RMSE less than 30 mol/(km$^2$·hr). We demonstrate
that the simultaneous update of NO$_x$ emission and concentrations outperforms the approach of updating emissions only. We recommend carrying out meteorology assimilations to stabilize from the initial transport errors before starting the emission inversion. Last but not least our results show that the information of wind *uncertainties* is not important for NO$_x$ emission estimation. Chemical assimilations of NO$_2$ only with the meteorology extracted from an hourly weather assimilation analysis would perform as well as the joint assimilation of meteorology and chemistry.


# 1 Introduction

Weather and climate act in concert with emissions to establish the concentrations of chemicals and aerosols in the boundary layer. To improve forecasts of exposure and understand the factors that affect public health the productivity of agriculture and animal husbandry, we require accurate models of both emissions and the boundary layer meteorology to define the surface layer concentrations that determine the exposure of humans, animals and plants. There remain substantial uncertainties in even the best models of emissions and even more so in the best models of boundary layer dynamics (for example, Hu et al., 2010). Current uncertainties in the surface $NO_2$ emission inventories in the U.S. are thought to be of order 50% (Krotkov et al., 2016; Travis et al., 2016). Comparable uncertainties affect estimates of the planetary boundary layer (PBL) height and mixing rates that redistribute emissions from the surface (Kretschmer et al., 2012, 2014; Lauvaux and Davis, 2014).

Over the last decade, data assimilation techniques have been used to constrain model forecasts and reanalyses of atmospheric constituents (Arellano Jr. et al., 2007; Edwards et al., 2009; Claeyman et al., 2011; Lahoz et al., 2012; Pagowski and Grell, 2012; Bowman, 2013; Gaubert et al., 2014; Hache et al., 2014; Saide et al., 2014; Zoogman et al., 2014; Barré et al., 2015; Bousserez et al., 2016; Mizzi et al., 2016). Assimilation of chemicals can be extended to optimize model inputs, such as emissions thereby providing insight into how to improve the processes that govern the model performance (Elbern et al., 2007; Barbu et al., 2009; Chatterjee et al., 2012; Miyazaki et al., 2012b; Koohkan et al., 2013; Yumimoto, 2013; Cui et al., 2015; Guerrette and Henze, 2015; Turner et al., 2015).

To date most efforts to incorporate satellite remote sensing in data assimilation have focused on long-lived chemicals such as CO, $CH_4$ or $CO_2$ and regional and continental scale aspects of emissions. Processes that govern variability of emissions within an urban center require new approaches that use high spatial and temporal resolution models and observations. $NO_2$ has a lifetime of only a few hours and thus exhibits concentration changes that are substantial on spatial scales of 50-75 km. Observations of variations in $NO_2$ are thus uniquely suited to study emissions and meteorology at the scales of cities. Averaged measurements of $NO_2$ have been shown to be promising for evaluation of absolute emissions and trends (Russell et al., 2012; Miyazaki et al., 2016) as well as providing information on the coupling of boundary layer winds to chemical lifetime (Beirle et al., 2011; Valin et al., 2013). Current space based instruments have resolution that is too low to provide direct information on lifetimes and emissions from a single overpass. Instead, analyses have focused on averages of the data that wash out some of the key details about emission location and chemical lifetime.

New instruments with spatial resolution of a few kilometers will soon change that situation. The TROPOspheric Monitoring Instrument (TROPOMI, launch date of 2017) will be the first to provide spatial resolution sufficient to observe these $NO_2$ changes on a single overpass. TROPOMI will view the atmosphere from low earth orbit and provide one image per day. We also anticipate the launch of three geostationary satellites, the Geostationary Environmental Monitoring Spectrometer (GEMS), the Tropospheric Emissions: Monitoring of POllution (TEMPO) and Sentinel-4, which will provide observations at higher temporal resolution with hourly repeat at locations in Asia, North America and Europe, respectively (Zoogman et al.,

2017). The spatial resolution of these new low earth orbit (LEO) and Geostationary (GEO) instruments will be sufficient to provide ~10 samples within the advection distance that is determined by the chemical lifetime of $NO_2$. This dense sampling will permit characterization of multi-exponential or non-exponential behavior where current analyses are typically forced to assume single exponential decay. To take full advantage of these measurements within a data assimilation system, we will

need to model the $NO_2$ column at similar spatial resolution. This is both because the spatial scales of important variation in atmospheric plumes are on the order of 4 km and because of the steep non-linearity in the lifetime of $NO_2$ as a function of the $NO_2$ concentration. For example, biases of 34% (3.3 to $5.0\times10^{15}$ molecules/cm$^2$) are found in the modeled averaged $NO_2$ column over Los Angeles at resolutions of 96 km compared to 12 km. For a point source, such as a power plant, model convergence is observed only at a grid resolution of 4 km or smaller (Valin et al., 2011).

In this study, we describe a high spatial and temporal resolution chemical transport ensemble data assimilation system with joint assimilation of meteorology and chemistry to adjust $NO_x$ emissions on scales consistent with the temporal scale of $NO_x$ evolution. We use that forecast/assimilation system to investigate the factors that influence the capability of TEMPO $NO_2$ observations to accurately constrain $NO_x$ emissions. Our long-term goal is to estimate hour-to-hour variations in $NO_x$ emissions at the scale of model grid point resolution (3 km) and to use these variations to understand the processes

controlling the emissions. The remainder of this paper is organized as follows: In section 2, we describe the forecast/data assimilation system, the system setup, observations, and the TEMPO $NO_2$ simulator – the simulation of column $NO_2$ that would be observed by TEMPO. In section 3, we describe the experimental design including a series of assimilation experiments that guide optimization of the emission estimation performance.   In section 4, we assess the performance of meteorology and chemistry assimilation. We then discuss the results and provide insight into the potential accuracy of $NO_2$

emission fields derived from geostationary $NO_2$ observations.  We conclude in section 5.

## 2 The data assimilation system

The forecast/data assimilation system used here is WRF-Chem/DART as described by Mizzi et al. (2016). It consists of the following elements: the forecast model, the assimilation engine, and observations of meteorological and chemical states to be assimilated.

**2.1 WRF-Chem model description**

The core meteorological and chemical forecast model is the regional online chemical transport model WRF-Chem v3.4.1 (www2.acd.ucar.edu/wrf-chem). The model domain is a one-way nest with an outer domain of 12 km resolution covering western North America and an inner domain of 3 km resolution focused on the city of Denver, CO (Figure 1). The 3 km domain is 660 km by 840 km. The model has 30 vertical levels between the surface and an upper boundary of 100 mb and 10

levels within the boundary layer (~1.5 km). Simulations of meteorology on the outer domain are initialized and constrained at the lateral boundary by North American Regional Reanalysis (NARR) data from National Centers for Environmental

Prediction (NCEP). The NARR data have a native horizontal resolution of 32 km with 45 pressure levels and 3 h temporal resolution. We use the global chemical model output from MOZART to initialize the chemical simulation on the outer domain and to provide the chemical boundary condition. After a spin-up time of four days on the outer domain, the inner domain simulation is initialized and constrained through one-way nesting in both meteorology and chemistry.

Anthropogenic emissions for WRF-Chem are from the National Emission Inventory (NEI) 2011 Version 1 at native $4\times4$ km$^2$ resolution. The NEI 2011 provides hourly-varying emission for a typical weekday in summertime. The emissions do not vary from day to day. Biogenic emissions are calculated online with the simulation results by Model of Emissions of Gases and Aerosols from Nature (MEGAN). Fire emissions are not included. We use the widely-used regional acid deposition model version 2 (RADM2) as the gas phase chemical mechanism (Stockwell et al., 1990). There are 59 species and 157

reactions to represent both inorganic and organic chemical reactions under tropospheric conditions. It includes the chemical losses of $NO_x$ through reaction with OH radical to form nitric acid, and other $NO_x$ sinks as peroxyacyl nitrates and alkyl nitrate. RADM2 predicts peak $NO_2$ concentrations and the timing that the peak occurs very well when the mechanism is tested against the representative environmental chamber experiments.

## 2.2 DART ensemble assimilation system

WRF-Chem/DART (data assimilation research testbed) is a regional multivariate data assimilation system developed by the National Center for Atmospheric Research (NCAR) to analyze meteorological variables and chemical variables simultaneously (Mizzi et al., 2016). We use the Ensemble Adjustment Kalman Filter (EAKF) in DART to analyze the states with an ensemble size of 30. Details of the EAKF algorithm and its implementation in DART are documented in (Anderson, 2001; Anderson and Collins, 2007; Anderson et al., 2009). In this study the system is extended to assimilate synthetic

TEMPO $NO_2$ column observations. As emissions are not prognostic variables of the forecast model, we implement a state augmentation approach to include emissions in the state variables (Aksoy, 2006). The chemical state variables include the $NO_2$ concentration, $NO_x$ emissions. The meteorological state variables are U, V, W, T, QVAPOR, QCLOUD, QRAIN, QICE, QSNOW (MU and PH are used in vertical coordinate transforms, T2, Q2, U10, V10, PSFC used for surface data assimilation forward operators.) based on the settings used in meteorology data assimilation (Romine et al., 2013). Adaptive

spatially and temporally varying inflation is applied to the prior state to assist in maintaining ensemble spread. We summarize the DART configuration details in Table 2.

### 2.2.1 Spatial localization

In ensemble methods the correlations among spatially remote variables in the prior ensemble are regarded as spurious correlations due to the small ensemble size (30). To compensate for this under-sampling issue, spatial localization is

introduced to reduce the prior correlations based on the distance between the observed/modeled state variables (Houtekamer and Mitchell 2001). In this study, we apply the fifth-order distance-dependent Gaspari and Cohn (GC) function (Gaspari et al., 1999) to reduce the spurious impact of observations on spatially remote state variables. The scaling distance in the GC

function is defined by a half-width parameter, two times which is the distance where the GC function goes to zero. With a data assimilation window of one hour and a maximum wind speed of 3~5 m/s, an observation of column $NO_2$ contains information about emissions that occurred during the last hour within 10 km. We use the half-width distance in spatial localization as 10 km and demonstrate this as the optimal value based on sensitivity experiments with localization distance of 5 km, 10 km, 20 km and 50 km. Because of the high density of TEMPO $NO_2$ observations ($2 \times 4.5$ km$^2$), the update of chemical state variables is mostly determined by the local observations.

### 2.2.2 Variable localization

Similar to the concept of spatial localization, variable localization techniques have been introduced (Arellano Jr. et al., 2007) to reduce spurious correlations among observations and different types of state variables. For example, for $CO_2$ flux estimation, Kang et al. (2011) showed that the performance of data assimilation using a variable localization that zeros out the prior error covariance between meteorological variables and $CO_2$ flux is better than using a standard full covariance approach. Here we isolate the influence of meteorological observations on chemical variables and vice versa. In our next paper we will discuss the potential benefit from assimilations of chemical observations to improve the meteorological fields.

### 2.3 Initial and boundary condition ensemble

We generate the initial chemical ensemble by adding the perturbations to the mean state of the fine domain forecast. In the ensemble method the generated ensemble should represent the error statistics of the initial guess of the model state (Evensen 2003). The correlation between perturbations of chemical state variables is modeled by a simple isotropic exponential decay function with a characteristic correlation length of 50 km. For meteorology ensemble, random perturbations were added to each member by sampling the NCEP background error covariance using WRF Data Assimilation System (http://www2.mmm.ucar.edu/wrf/users/wrfda) (Barker et al., 2012). The options in WRFDA setting is summarized in Table 3. The ensemble member lateral boundary condition perturbations are generated in a similar manner as the initial ensemble using the fixed-covariance perturbation technique. The boundary condition for the analysis time is adjusted to match the analysis from DART. The tendencies for the later times in the forecast are adjusted to match the change in the boundary condition for the analysis time.

### 2.4 Emission update scheme

By including emissions in the ensemble state vector, emissions are estimated as hourly evolving parameters. Estimation of time-evolving emissions using data assimilation is first presented in carbon fluxes estimation (Kang et al., 2011, 2012). Such an approach provides emission information more than a monthly mean, or an average for a specific time period. For $NO_x$ emission estimation problem, emissions in cities show significant variation from urban to suburban. The observed columns show strong spatial variation dominated by emission hotspot as a result of the short lifetime. Combined with high-resolution

observations, the goal in this work is to constrain a time-static emission error as we perturb emissions of each hour in the same way.

A challenge for updating the emissions in the augmented state vector is the absence of an emission forecast model to evolve the emission variables forward in time. The bottom-up inventory to be optimized provides hourly-resolved emissions for each model grid point. Instead of treating the emission variables of each hour at a specific location as independent parameters, we update the emission scaling factors at each assimilation cycle. For a given model grid point, we define the emissions of truth ($e_i^t$), prior ($e_i^{prior}$) and posterior ($e_i^{post}$) at the ith cycle. Since we start the assimilation with 70% of true emissions, we have $e_i^{prior} = 0.7 e_i^t$, for each cycle. After one assimilation cycle, we compute the scaling factor ($S_i$) as follows: $S_i = e_i^{post}/e_i^{prior}$. Then we update the prior emissions at the next cycle as $e_{i+1}^{prior} = S_i \cdot e_i^{prior}$. This prescription enables us to derive spatial 2-D emission scaling factors which play the role of an emission forecast model.

## 2.5 Synthetic meteorological and chemical observations

Observations assimilated in the system include meteorological observations and $NO_2$ trace gas retrievals from the TEMPO OSSE. For meteorological observations, we assimilated synthetic observations of temperature, wind and humidity from the NCEP Meteorological Assimilation Data Ingest System (MADIS) (https://madis.noaa.gov/). MADIS is a meteorological observational database and data delivery system that provides observations that cover the globe. MADIS ingests data from NOAA data sources and non-NOAA providers, decodes the data then encodes all of the observational data into a common format with uniform observational units and time stamps. For wind observations, the assimilated observation types include standard aviation routine weather report (METAR), wind profilers, aircraft-based observations (ACARS), national mesonet data and satellite data. Among them the mesonet wind data is the most abundant with ~1000 observations located in the mapping domain in Figure 2. The observation errors are the default values from the DART facility that are defined based on NCEP statistics (Romine et al., 2013)

The GEOstationary Coastal and Air Pollution Events (GEO-CAPE) mission (Fishman et al., 2012) aims at improving our understanding of both coastal ecosystems and air-quality from regional to continental scales. As the first phase of the GEO-CAPE implementation, TEMPO (Zoogman et al., 2017), launch date circa 2019, will provide hourly measurements of $NO_2$, HCHO, tropospheric ozone, aerosols, and cloud parameters during the daytime. TEMPO will measure solar back scattered light in the UV-Vis spectral range. Implemented on a geostationary platform, TEMPO retrievals will achieve hourly observations of $NO_2$ vertical column density (VCD) at a native spatial resolution of 2×4.5 km during the day-lit period. TEMPO's high spatiotemporal resolution will allow a more detailed assessment of emission inventories, e.g. urban scale and large power plant $NO_2$ emissions and mobile emissions that show significant spatial and temporal variations due to urban transit patterns, than is possible with existing LEO observations.

As the TEMPO has not been launched yet, we generate synthetic TEMPO $NO_2$ observations by simulating the instrument's observing characteristics. We carried out a model run, i.e. a forward integration of WRF-Chem for the period from July 2[nd]

to July 7$^{th}$ 2014 with NO$_2$ emissions specified by NEI 2011 ('truth'). We calculate a layer dependent Box-Air Mass Factor (BAMF) representing the sensitivity of the retrieved NO$_2$ in a specific layer to the true value in the atmosphere. The BAMF of NO$_2$, as an optically thin absorber, is a vector and determines the measurement sensitivity to NO$_2$ molecules at 35 pressure levels. In the calculation of BAMFs, we follow the latest version of the NASA standard product retrieval (Level 2, Version 2.1, Collection 3) algorithm (Bucsela et al., 2013) assuming the TEMPO measurement has similar characteristics to OMI. We assume clear-sky conditions for all observing scenes. Cloudy-sky scenes affect only the number of observations available as the cloudy scenes are usually discarded in the data filtering process. The elements of the BAMF vector are computed as a function of solar zenith angle (SZA), viewing zenith angle (VZA), relative azimuth angle (RAA), terrain reflectivity (Rt), terrain pressure (Pt), atmospheric pressure level, (p) and the NO$_2$ profile (Bucsela et al., 2013). The viewing parameters are computed by simulating viewing geometry based on the location of ground pixels in relation to the observing instrument. The geometry related parameters (SZA, VZA and RAA) are computed hourly for each TEMPO observation using Matlab functions sun_position.m and geodetic2aer.m with inputs of the location and time of each TEMPO observation, and the location (36.5°N, 100°W) and altitude (35,786 km) of the TEMPO sensor. The terrain reflectivity and terrain pressure are sampled from the WRF-Chem nature run (NR, see section 3) for each TEMPO pixel. All the parameters have an hourly frequency consistent with the TEMPO temporal observing pattern. Consequently, the NO$_2$ profile with high-spatial-temporal resolution captures the diurnal variation of NO$_2$ and its urban-rural contrast. This contrast is essential to accurate interpretation of the measured spectrum (Russell et al., 2011; Laughner et al., 2016).

To generate synthetic TEMPO data, the modeled 3-D concentration fields from the NR are sampled in as similar a manner to the planned TEMPO measurements as the transport model permits: using the computed BAMF vertically; hourly frequency; 2×4.5 km nadir resolution and variations following the Earth's curvature horizontally. Figure 2 shows an example of the spatial distribution of TEMPO data over Denver, CO.

We describe the observation error as a relative value ($\sigma_{rel}$) and a random draw from a Gaussian distribution to avoid using a fixed value. The magnitude of the mean uncertainty of the NO$_2$ column is different between clean and polluted areas (Boersma et al., 2004). We follow their categorization of clean versus polluted regions and summarize the mean and standard deviation of a Gaussian distribution for each scenario in Table 4. For polluted regions, we give mean uncertainty of 7.5%, which is lower than the 35% minimum in the OMI NO$_2$ retrievals. First, most of these errors are systematic affecting comparison of different cities but have smaller variation across a single, small area scene of observations. Second, a relatively lower observation error improves the efficiency of data assimilation and helps to examine the sensitivity to other parameters. Finally, as TEMPO is expected to be operational no sooner than 2018, it is reasonable to expect the retrieval error dominated by air mass factor (AMF) in polluted region will be reduced as a result of future improvements in AMF simulation (Laughner et al., 2016). The synthetic observations assimilated are obtained by sampling the NR using the TEMPO observation simulator and adding observation error as $y^{obs} = N(y^{tr}, \sigma^2)$, where $y^{tr}$ is the TEMPO NO$_2$ observations sampled from the truth, and $\sigma$ is the observation error standard deviation computed as $\sigma = y^{tr} \cdot \sigma_{rel}$.

# 3 Assimilation experiments

We begin by performing observing system simulation experiments (OSSEs) in the context of a perfect model. The original NEI 2011 is used as the emission input for the NR without any emission perturbation. We consider the NR as the true atmosphere and sample meteorological and $NO_2$ observations from the NR. The control run (CR) is a parallel model calculation to the NR and suffers from imperfect model input and parametrization. The differences between the NR and the CR in this study are the emission inputs and the initial conditions for the meteorology. We begin by creating a NR and a CR simulation on the outer domain of 12 km resolution (d01) without assimilating observations using a simulation setup as described above in section 2.1. We impose a difference to the CR by using emissions in the CR that are scaled to be 70% of the NR emissions. We apply the identical forecast model (WRF-Chem) for both the NR and the CR to isolate the behavior of the ensemble filter algorithm from the influence of the model errors. Then the NR and the CR on the inner domain of 3km (d02) are initialized from the corresponding d01 simulations respectively on 06:00 local time (LT) on July $2^{nd}$ 2014. At the time of initialization, the NR and CR on d02 share the same meteorological fields and differ in $NO_x$ concentrations due to different emission inputs. Our next step is to generate a 30-member ensemble from the CR. We use WRFDA to generate an ensemble in meteorological variables (Barker et al., 2012). For chemical states, we give an ensemble in $NO_x$ emissions and concentrations using the method described above in section 2.3. The forecast of the CR ensemble is the prior estimate of the states and will be combined with the observations in the assimilation cycle to yield the posterior states. By comparing the posterior emissions with the "true emission", we evaluate the data assimilation performance. We run assimilation experiments from 10:00 LT 2014/07/02 to 18:00 LT 2014/07/05 with an assimilation window of one hour. We assimilate ~20,000 weather observations in each assimilation window and ~9,000 TEMPO $NO_2$ column observations in each daytime assimilation window.

We design a series of experiments to explore the optimal approach to estimate $NO_x$ emissions as shown in Table 1. In all experimental runs, we bias the CR initial emissions to be 30% below the reference model and examine the ability of the assimilation to recover the reference emissions. First, a reference assimilation run (REF) is conducted without including the meteorological ensemble so that the NR and CR ensembles have identical meteorological simulations. This shows the best case scenario to constrain emissions assuming no errors associated with meteorology. In practice, the modeled meteorology is different from the true atmosphere due to errors in the model initial conditions, parameterizations and resolutions. In a real case labelled as ENS, we initialize an ensemble in both meteorology initial conditions and $NO_x$ emissions in which both weather observations and TEMPO $NO_2$ columns are assimilated. In ENS.1 the CR ensemble is generated by adding perturbations to the CR mean state, so the CR ensemble mean meteorology is very close to the NR because CR and NR differ in $NO_x$ emissions only. For the chemistry part, the assimilated TEMPO $NO_2$ observations will update both $NO_2$ concentration and $NO_x$ emissions every hour. In ENS.2 we only allow $NO_2$ observations to update $NO_x$ emissions and keep the meteorology assimilation the same as ENS.1. By comparing ENS.1 and ENS.2 we can determine whether or not concentrations should be updated when observations are assimilated to constrain unobserved emissions. In ENS.3, we use

the meteorology of the next day to initialize the CR ensemble so that there is some difference between the CR ensemble mean and the NR in the meteorology. To be specific, the CR meteorology ensemble on 2014/07/03 9:00 LT is used as the CR ensemble on 2014/07/02 9:00 LT. This is to mimic our imperfect knowledge about the atmosphere state and its uncertainty in reality. ENS.1 and ENS.3 differ only in the meteorology initial ensemble and by comparing these two runs, we can see the sensitivity of the NO$_2$ assimilation to the meteorology initialization. Our final experiment REA mimics a general approach in chemistry-only data assimilation where the meteorology is extracted from an existing reanalysis. REA reinitializes the meteorological state every hour with the best estimate of meteorological states generated by ENS.1. As a result, REA has a single run of meteorology but still an ensemble of NO$_2$ emissions and concentrations which allow it to assimilate TEMPO NO$_2$ observations. Same as ENS.1, REA updates emissions and concentrations simultaneously.

## 4 Results

We evaluate the assimilation result by comparing with the NR states. We calculate the root mean square errors (RMSE) of observed quantities by $\sqrt{\sum_i^n (y_i^m - y_i^t)^2/n}$, where $y_i^m$ and $y_i^t$ are the model and true values for the ith observation respectively, and n is the total number of observations of interest. We also calculate the RMSE of model states by $\sqrt{\sum_i^n (x_i^m - x_i^t)^2/n}$, where $x_i^m$ and $x_i^t$ are the model and true states at the ith model grid point respectively, and n is the total number of grid points of interest. For the wind variable, the grid points of interest are all the points located within a sub-model space as shown in Figure 2, containing the lowest 10 model levels vertically. Because NO$_x$ is located mostly in the boundary layer, the NO$_2$ transport error is determined by the meteorological errors in the lowest 10 model levels. For NO$_x$ emission variables, the grid points of interest are categorized as emission points with emissions greater than 50 mol/(km$^2$·hr). Our analysis does not include emissions below 50 mol/(km$^2$·hr) because the observations over such low emission regions have large uncertainty and are not constrained. We also analyze the uncertainty of the prior and posterior estimates. The uncertainty is expressed by the 1-σ standard deviation of the ensemble.

### 4.1 Wind assimilation

The success of ensemble-based assimilation relies on how well the ensemble system represents the uncertainty. One way to test the success of an OSSE is to compare the RMSE computed with respect to the "true" observations with the ensemble spread directly. Figure 3 shows the evolution of the RMSE and spread for mesonet observations of zonal wind for ENS.1 and ENS.3. Overall, for each experiment the variation and magnitude of prior ensemble spread are similar to those of the prior RMSE, indicating that the ensemble develops a good amount of spread for the success of OSSE.

We find the errors in the observation space of mesonet winds are reduced by 50% on average from the prior to the posterior. The prior wind RMSE shows peaks in the afternoon and this results in the most significant error reduction over the time period. The posterior wind RMSE shows a temporal average of 0.39 m/s and 0.47 m/s in ENS.1 and ENS.3 respectively. Because ENS.1 is initialized with a meteorology ensemble with its mean close to the truth, the wind RMSE on the first day is

low and gradually grows to about 1 m/s. In contrast, the prior wind RMSE in ENS.3 is as high as 2 m/s on the first day as a result of using an initial meteorology ensemble that is very different from the truth. The wind RMSE evolution in the two experiments becomes very similar after the afternoon of 2014/07/04. We conclude that the ensemble wind assimilation system performance is independent of the initialization approach after the first day.

**4.2 TEMPO NO$_2$ assimilation**

We assimilate hourly TEMPO NO$_2$ column observations and take their difference with the modeled column to correct the predicted NO$_2$. Figure 4 shows the TEMPO NO$_2$ column RMSE evolution for all experiments. With perfect knowledge of meteorology, REF shows significant reduction in TEMPO NO$_2$ RMSE in the first three update cycles, and succeeds in recovering the true emissions (Figure 5). The prior TEMPO NO$_2$ RMSE in the last three days varies below $3 \times 10^{14}$

molecules/cm$^2$ as a result of perfect NO$_2$ transport and improved emissions. This ideal case with the assumption of perfect meteorology sets the upper limit of error reduction in NO$_2$ concentrations by assimilating the TEMPO NO$_2$ observations. Compared with REF, ENS.1 shows prior TEMPO NO$_2$ RMSE of $5 \sim 10 \times 10^{14}$ molecules/cm$^2$ due to the errors in NO$_2$ transport and emissions. By assimilating NO$_2$ observations, the TEMPO NO$_2$ RMSE is reduced by more than 50% from the prior to the posterior indicating the potential of TEMPO NO$_2$ observations to improve the modeled atmospheric NO$_2$

composition for the chemical reanalysis product. Without updating the NO$_2$ concentrations in ENS.2, there is no reduction in the TEMPO NO$_2$ RMSE as expected. We find the TEMPO NO$_2$ RMSE varies above $1 \times 10^{15}$ molecules/cm$^2$, being the largest among all experiments because the emission estimations show very poor results (shown in section 4.3). The TEMPO NO$_2$ RMSE development in ENS.3 is very similar to ENS.1 except for the first day when ENS.3 shows higher errors in the wind field, which contribute to the NO$_2$ transport errors. We find the NO$_2$ forecast using a single meteorology field in REA is very

similar to the ensemble NO$_2$ forecast in ENS.1. This is because there is very little difference between the one-hour meteorology forecast and the ensemble forecast. In addition, the emission estimation results are also very similar. This is different from the previous study on CO$_2$ forecast which shows that for a 6-hour forecast, the CO$_2$ transport driven by a single meteorological field has weaker vertical mixing and stronger CO$_2$ vertical gradient when compared to the mean of the ensemble CO$_2$ forecasts initialized by the ensemble meteorological field (Liu et al., 2011).

We compare the TEMPO NO$_2$ column spread in REF and ENS.1 in Figure 5. For both experiments, the prior NO$_2$ column spread varies at the same magnitude with the prior RMSE (Figure 4), indicating good quality of the NO$_2$ ensemble spread. The NO$_2$ forecast uncertainty represented by the NO$_2$ ensemble spread results from the uncertainties in NO$_2$ transport and emissions when the uncertainties in chemical production and removal processes are not included in this study. The uncertainties in NO$_2$ transport are determined by the prior wind ensemble spread, which shows peaks in the afternoon and

stays as low as $\sim 0.5$ m/s at other times for zonal wind (Figure 3). The prior NO$_x$ emission uncertainties are 60% after inflation (Figure 6). Under these circumstances, the mean prior TEMPO NO$_2$ column spread is $4.55 \times 10^{14}$ molecules/cm$^2$ in REF which does not include NO$_2$ transport uncertainties, and is $7.03 \times 10^{14}$ molecules/cm$^2$ in ENS.1 which takes uncertainties in transport and emissions into account. So the NO$_2$ transport contributes to 35% of the total NO$_2$ forecast uncertainties in

our assimilation setup. The TEMPO $NO_2$ column spread in REF is very stable because it is determined by the constant emission spread of 60%. ENS.1 shows fluctuations in the evolution of TEMPO $NO_2$ column spread which corresponds to the wind spread variation with increasing spread in the afternoon.

### 4.3 $NO_x$ emission estimation

In our emission update scheme, the TEMPO $NO_2$ observations at time t are assimilated to generate a scaling factor for emissions at t-1 hour. This scaling factor is used to correct the prior emission at time t. In this way, the model-observation difference in the $NO_2$ column will correct the emission of an hour ago instead of the current emission. This approach is reasonable because errors in $NO_2$ concentration result from errors in previous emissions. Considering the short $NO_2$ lifetime of three hours in summer daytime, emissions from the previous hour have a large contribution to the $NO_2$ total mass at the current time.

We show the time evolution of the averaged city emissions for all experiments in Figure 6. For all experiments, the posterior emission ensemble spread is reduced compared to the prior spread, suggesting the effectiveness of assimilated $NO_2$ columns in constraining the emission uncertainties. We should mention that we ignore the emission correction of the first assimilation cycle. We notice that the first update produces a significant over-correction on emission because of the accumulated underestimation of $NO_2$ from the previous time. By neglecting the first update, the prior emission ensemble mean of the second cycle is still 70% of the truth. During the nighttime when TEMPO observations are not available, we calculate the ratio of posterior ensemble mean to the truth in the last cycle of daytime and use this together with the nighttime true emissions to derive the ensemble mean for the nighttime emissions. The prior and posterior emission ensemble of each nighttime hour is the same.

Not surprisingly, under the condition of perfect knowledge in meteorological fields, assimilating TEMPO $NO_2$ observations successfully improves the emissions within the first few updates. The estimated emissions agree well with the true emissions throughout the assimilation period. This demonstrates the capability of a geostationary $NO_2$ column observing system to constrain city-scale emissions and the reliability of the ensemble-based assimilation method to project the observed information to emissions.

We find the errors in estimated emissions correlate with the wind errors. In ENS.1, the posterior emission is corrected to the truth at the second cycle and stays close to the truth throughout the first day. The good performance on the first day benefits from an initial meteorology ensemble with its mean close to the truth. For the following three days, the emission estimates succeed in recovering the true emissions during the morning and show deviations from the truth in the afternoon as a result of the increased error in boundary layer winds. Figure 7 shows the dependence of errors in the inverted emissions to the prior wind RMSE. The emission errors show high sensitivity to the wind errors with the slope of the regression line of 32.5 mol·km$^{-2}$hr$^{-1}$/(m·s$^{-1}$). With the RMSE of model predicted wind vectors of 1 m/s, the errors in the estimated emissions are 30 mol·km$^{-2}$hr$^{-1}$ on average. For the daytime cycles, the prior emission ensemble spread after inflation is approximately 60%

and is reduced by more than half after assimilation (Figure 6). Even though the posterior ensemble mean does not match with the truth in the afternoon, the truth falls within the range of the posterior ensemble spread with a few exceptions.

We find the simultaneous update of emission and concentration performs better than the emission update only scheme with an hourly assimilation window. ENS.2 is a parallel assimilation run with ENS.1 but updates emissions only. As shown in Figure 6, the estimated emissions have very large differences from the truth and the posterior ensemble spread does not cover the truth. For example, at 10:00 July 3$^{rd}$, the posterior ensemble mean (red) is very close to the truth. As a result of this, we have a very good prior ensemble estimate (black) at 11:00. However, the posterior emission at 11:00 is largely underestimated compared with the truth. This is because the posterior emissions from 7:00 to 9:00 are overestimated which results in overestimated $NO_2$ concentrations at 10:00 and 11:00. As a result, even though the prior emissions from 10:00 to 11:00 are good, the model still overestimates $NO_2$ at 11:00 due to the $NO_2$ overestimation at 10:00. Without updating the concentrations, the observed differences in $NO_2$ columns are dominated by the $NO_2$ concentration errors of an hour ago and should not be attributed to the emissions.

We also find that the emission estimation should start after the meteorology assimilation becomes stable. As a comparison to ENS.1, ENS.3 is initialized with a meteorology ensemble that is very different from the truth. On the first day, the prior wind RMSE varies from 1 m/s to 2 m/s (Figure 3) and leads to enhanced $NO_2$ transport errors. As a result, the emission estimations are not successful for the first day. After the afternoon of the second day (07/03), the wind RMSE evolution is similar between ENS.1 and ENS.3 and as a result, the emission estimations perform in a similar way. We recommend carrying out meteorology assimilations to stabilize from the initial transport errors before assimilating chemical observations to constrain the emissions.

With an hourly re-initialization of meteorology, the $NO_2$ transport error statistics are not important in emission estimation in the current practice of a single meteorological field to transport $NO_2$. The emission estimation performance in REA is very similar with that in ENS.1 (Figure 6). This is because the difference in the one-hour $NO_2$ forecast driven by an ensemble meteorological field and a single ensemble mean field is very small. Though the wind uncertainties represented by the meteorological ensemble reach 1.5 m/s in the afternoon, our results show that the information of wind uncertainties is not important for $NO_x$ emission estimation.

Finally we examine the emission estimation performance in ENS.1 at the scale of the model grid (3 km). As shown in Figure 8, the true emission shows high spatial variation from city center to suburban as well as distinct point emission sources. In the example of the emission estimate at 9:00 am, the posterior emission recovers the truth very well with the posterior RMSE of 21.6 mol/(km$^2$·hr). In contrast, the emission estimate at 4:00 pm shows RMSE of 46.5 mol/(km$^2$·hr) due to relatively high wind errors. The posterior underestimates the emissions significantly all over the city except for the regional overestimation in the east. The emission hot spot of ~250 mol/(km$^2$·hr) in the city center is not fully represented in the posterior estimate.

## 5 Summary and conclusions

In this study, we demonstrate the ability to estimate $NO_x$ emissions by assimilating column $NO_2$ and meteorological observations into a data assimilation system comprised of regional CTM WRF-Chem and DART-EAKF. Such an ensemble-based data assimilation system is appealing for regional chemical transport studies as it allows the flexibility and efficiency to assimilate observations with various scales and of various chemical species. This is especially true, at present, in light of the availability of future long-term multi-species observations from satellite, such as GEO-CAPE and TROPOMI.

Previous work has shown that $NO_x$ concentrations and columns vary at fine scales necessitating high spatiotemporal resolution to make use of them in the assimilation. In the coupled chemical and meteorological data assimilation system, we apply an OSSE framework to estimate $NO_x$ emissions in Denver by jointly assimilating MADIS observation of meteorological variables as well as future TEMPO $NO_2$ columns. In meteorological assimilation we successfully reduced the posterior wind RMSE below 0.5 m/s in Denver to better represent the $NO_2$ transport. The prior wind RMSE and spread show peaks in the afternoon thus increasing the errors in $NO_2$ transport. We find that the meteorological uncertainties contribute 35% to the total $NO_2$ forecast uncertainties considering the emission uncertainties of 60%. Assimilation of TEMPO $NO_2$ columns reduces errors in the predicted $NO_2$ concentration by more than 50%, which demonstrates the potential of future geostationary observations to constrain $NO_2$ chemical weather.

One of the goals of this work is to investigate the optimal strategy to estimate $NO_x$ emissions. We test the upper limit of emission constraint from TEMPO $NO_2$ observations in an ideal case assuming no errors associated with the modeled meteorology. In the experiment of joint assimilation of meteorology and chemical $NO_2$, we find that the emission estimation is successful in the morning but degrades in the afternoon when the prior wind RMSE grows above 1 m/s. Considering the dependence of errors in estimated emissions on the wind forecast errors, we recommend to guarantee the accuracy in modeled wind with RMSE below 1 m/s for the success of chemical assimilation to infer emissions at the scale of our model grid. We show that the simultaneous update of $NO_x$ emissions and concentrations outperforms the approach of updating emissions only. We recommend carrying out meteorology assimilations to stabilize from the initial transport errors before starting the emission inversion. Last but not least our results show that the information of wind uncertainties is not important for $NO_x$ emission estimation. Assimilations of $NO_2$ only with the meteorology constrained from an hourly weather assimilation product would perform as well as the joint assimilation of meteorology and chemistry.

The emissions optimization scheme used in this study assumes no errors in the diurnal variation of emissions, which enables us to optimize the emission scaling factors with hourly frequency. For future work, if there is a large uncertainty in the temporal variability, the emissions at different time steps can be treated as independent state variables to allow the possibility of updating the prior temporal pattern of emissions. In our OSSE, we start with an ideal case of a priori emissions, which is spatially-uniform 70% of true emissions. Success in recovering emissions with a uniform scaling can't guarantee success in recovering emissions with spatially-varying perturbations. However, an approach that didn't work in the ideal case will fail in the realistic scenario with spatially-inhomogeneous emission perturbations. The degradation of emission estimation in the

ENS.2 and ENS.3 will still hold for correcting emission errors under more complex circumstances, such as spatially- or temporally- varying emission errors. For future work, further analysis is required to examine whether spatially-varying emission errors can be reduced under the condition of low wind RMSE.

Finally, we note that the methodology developed in this study can also be applied to emissions estimates of other chemical species or time-evolving parameters. For chemical species with longer chemical lifetimes, care should be taken in constraining the errors in transport and boundary conditions and preserving chemical covariance, as emission-induced forcing will be less dominant.

Acknowledgements. The authors gratefully acknowledge support from the NASA Grant NNX14AH046 and NNX15AE376. We thank N. Collins (NCAR/IMAGe) and T. Hoar (NCAR/IMAGe) for the assistance with DART. We would like to acknowledge high-performance computing support from Yellowstone (ark:/85065/d7wd3xhc) provided by NCAR's Computational and Information Systems Laboratory, sponsored by the National Science Foundation. We also thank the reviewers of this manuscript for their constructive suggestions.

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

**Table 1.** The experimental set up of each assimilation run. The three ensemble runs assimilate $NO_2$ observations every hour, and differ in treatment of meteorology forecast.

| Experiment | Met Assim | Chem Assim | note |
|---|---|---|---|
| REF | No | Yes | Truth meteorology |
| ENS.1 | Yes | Yes | Ensemble of meteorology and chemistry |
| ENS.2 | Yes | Yes | Only update emissions |
| ENS.3 | Yes | Yes | Initial meteorology ensemble is from the next day |
| REA | No | Yes | Using ensemble mean from ENS.1 |

**Table 2.** DART configurations.

| Parameter | Value |
|---|---|
| Filter type | EAKF |
| Adaptive inflation | 1.0, 0.6 (initial mean, spread) |
| Inflation damping | 0.9 |
| Adaptive localization threshold | 2000 |
| Localization type | Gaspari-Cohn |
| Horizontal localization half-width for meteorology (chemical) observation | 50 km (10 km) |
| Outlier threshold | 3.0 |
| Ensemble members | 30 |

**Table 3.** WRFDA configurations.

| Parameter | Value |
|---|---|
| cv_options | 3 (NCEP background error model) |
| je_factor (ensemble covariance weighting factor) | 1.0 |

**Table 4.** Relative observation uncertainty in synthetic TEMPO $NO_2$ column for each scenario.

| Type | $NO_2$ column | Gaussian distribution |
|---|---|---|
| Clean | $<0.3\times10^{15}$ molec. com$^{-2}$ | N (200%, 100%) |
| Polluted | $>=0.3\times10^{15}$ molec. com$^{-2}$ | N (7.5%, 2.5%) |

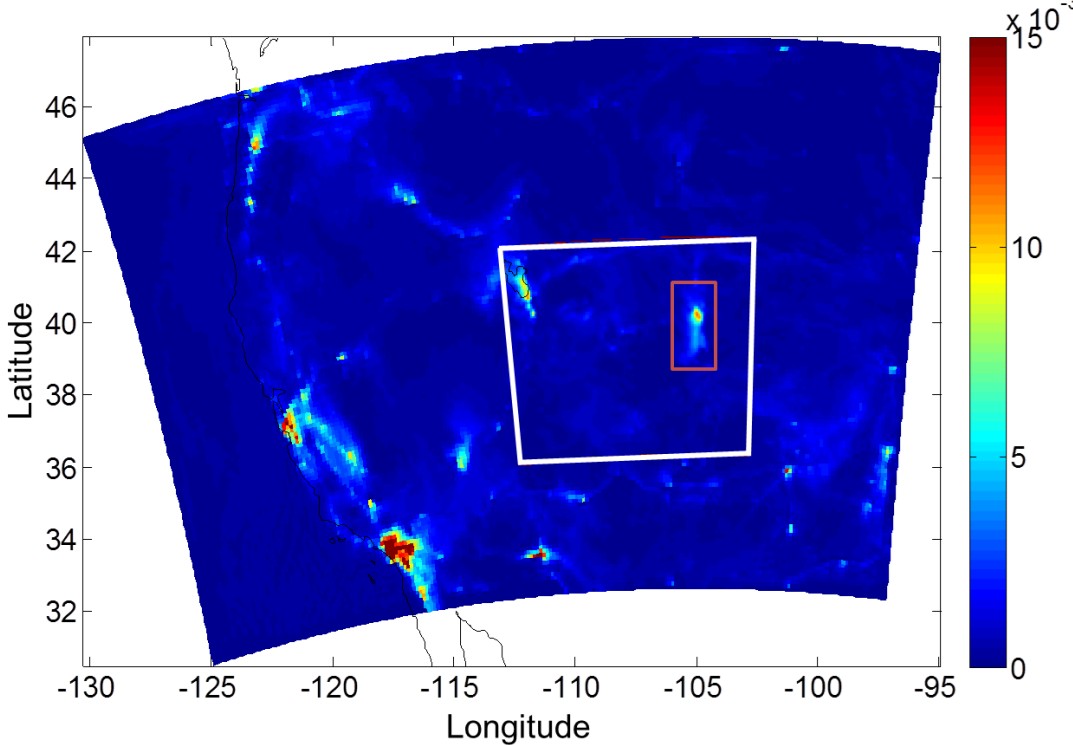

**Figure 1.** Model domain setup with 12 km outer domain and 3 km inner domain (white square). Data assimilation is performed on the inner domain. Meteorological observations on the inner domain are assimilated. TEMPO $NO_2$ observations inside the red rectangle are assimilated.

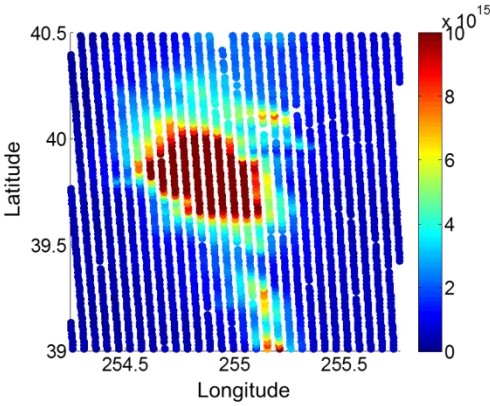

**Figure 2.** Example of synthetic TEMPO $NO_2$ column observations over Denver, CO at 17:00 LT July $2^{nd}$ in 2014.

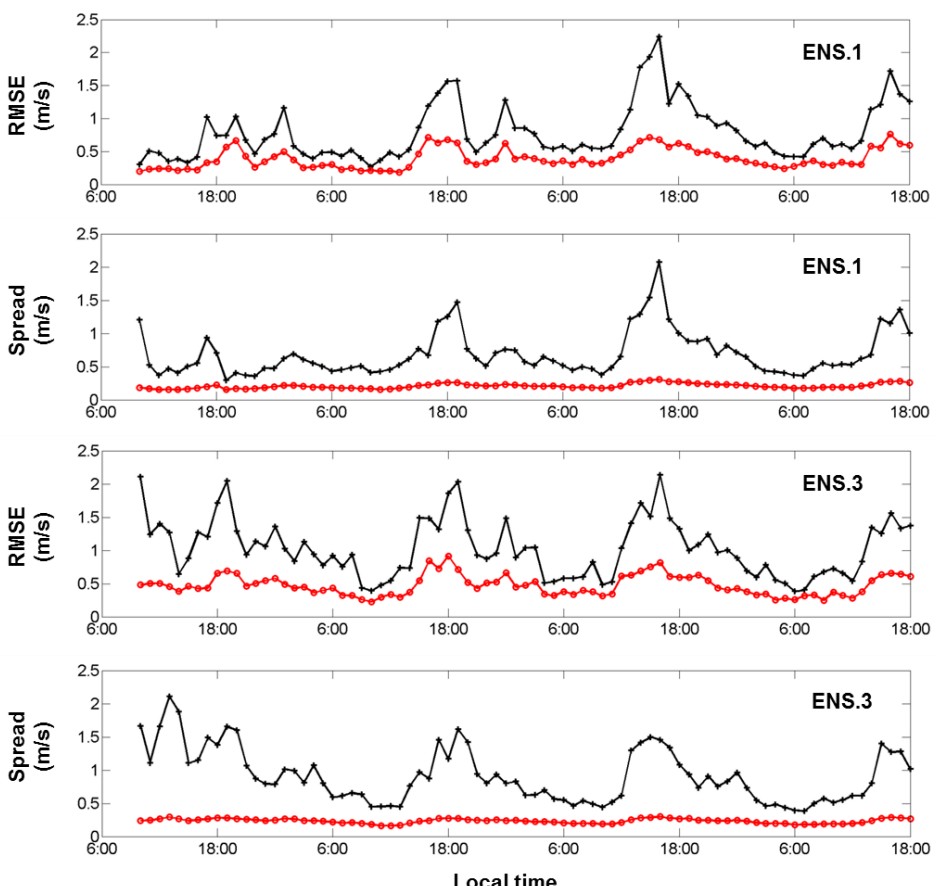

5  **Figure 3.** Time evolution of prior (black) and posterior (red) RMSE and spread of surface mesonet zonal wind observation in Denver from July 2$^{nd}$ 10:00 to 5$^{th}$ 18:00 for ENS.1 (top) and ENS.3 (bottom).

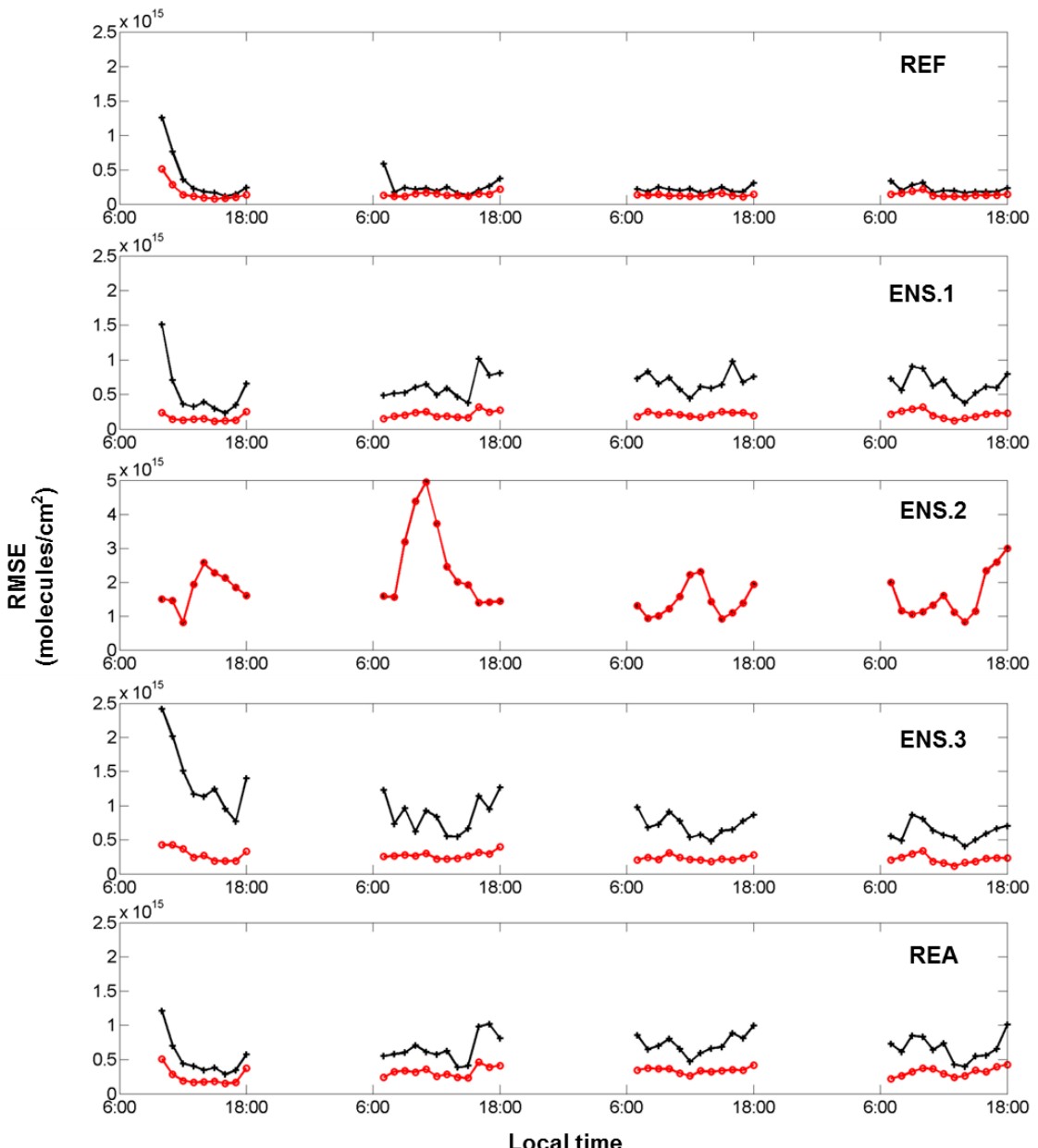

**Figure 4.** Time evolution of prior (black) and posterior (red) RMSE of Denver TEMPO NO$_2$ column observation from July 2$^{nd}$ 10:00 to 5$^{th}$ 18:00 for REF, ENS.1, ENS.2, ENS.3 and REA (from top to bottom).

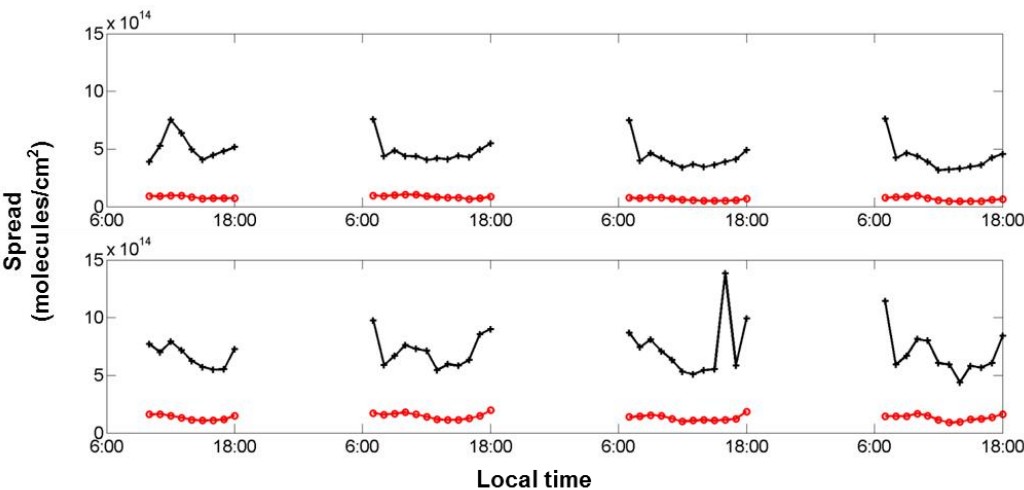

**Figure 5.** Time evolution of prior (black) and posterior (red) spread of Denver TEMPO $NO_2$ column observation from July 2[nd] 10:00 to 5[th] 18:00 for REF (top) and ENS.1 (bottom).

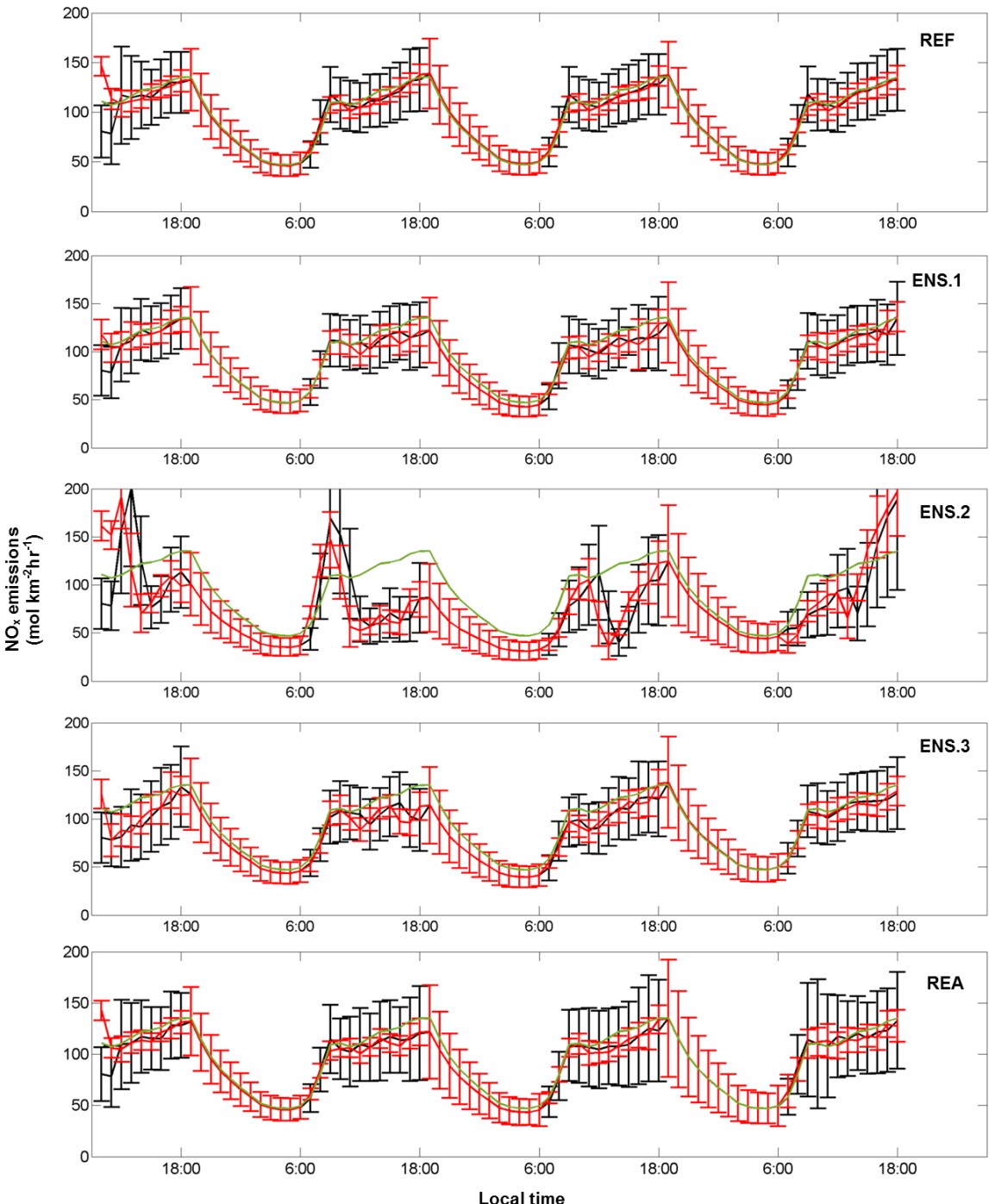

**Figure 6.** Time evolution of averaged Denver city emission of prior (black), posterior (red) and truth (green) for REF, ENS.1, ENS.2, ENS.3 and REA (from top to bottom).The error bar is defined by the ensemble spread and represents the uncertainty of the prior and posterior estimates.

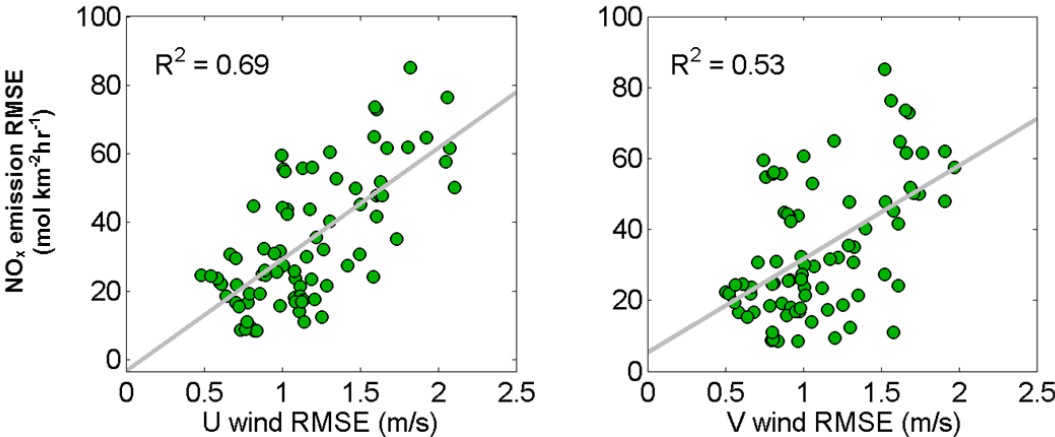

**Figure 7.** The scatter plot between the prior RMSE of boundary layer wind vectors and urban $NO_x$ emission posterior RMSE over four-day daytime assimilation time period in ENS.1.

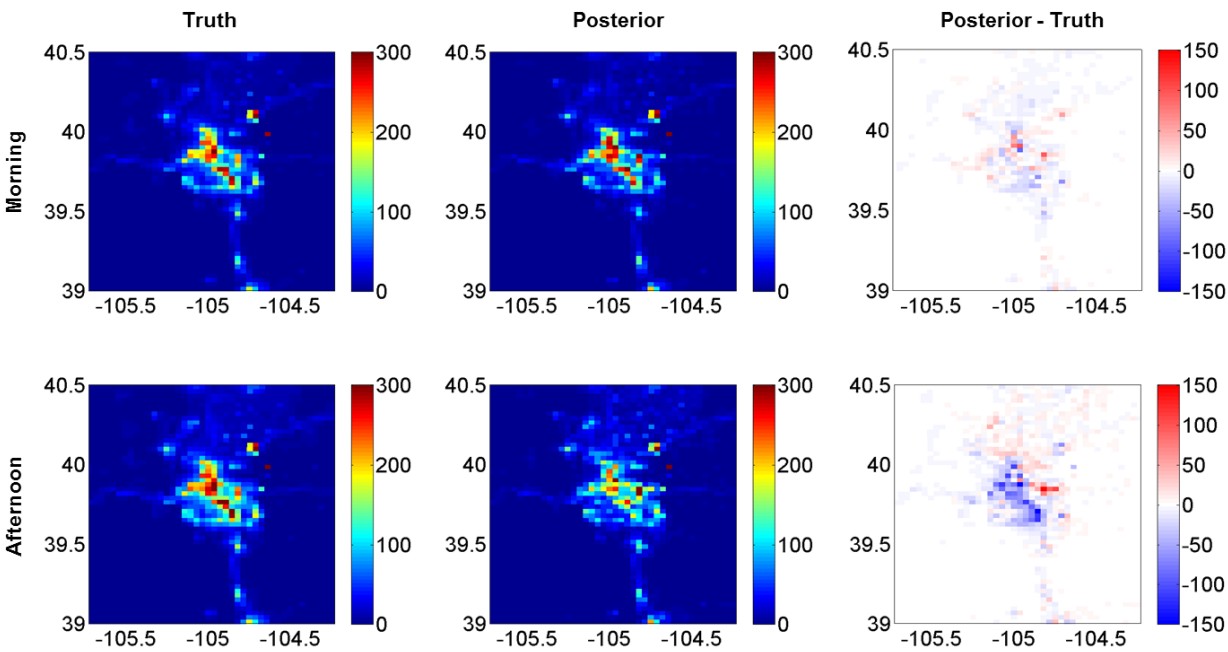

**Figure 8.** The emission estimation results in ENS.1 at 9:00 am (top) 4:00 pm (bottom) on July 3[rd] of truth, posterior and the difference between truth and posterior (from left to right). The unit is mol km[-2]hr[-1].