# Peer review of "Assimilation of satellite NO2 observations at high spatial resolution using OSSEs"

_Atmospheric Chemistry and Physics, 2016_

## Referee Comment (RC1) · Anonymous Referee #1 · 16 Oct 2016

In their paper, X. Liu and co-authors discuss the impact of future geostationary NO2 observations on air quality models and the potential of these observations to improve estimates of emissions.

General assessment:

The highlight of this paper is a documentation of the sensitivity of NOx emission estimates on the quality of the meteorological fields and the winds in particular. As shown, these issues are especially critical when the resolution of the observations and the models advances to the km scale. This sensitivity is demonstrated in a clear way, and to my opinion is an important aspect to keep in mind for the development of future regional analysis systems. As such, I am in favor of publishing these results in ACP.

The paper is well written, has a good introduction, but the overview of the method

and tools is very short and condensed. The implementation of the OSSE setup is not provided with enough detail for other groups to repeat the experiment, see my detailed comments below. I therefore would urge the authors to expand these parts of the paper.

In order to judge the importance of the meteorological uncertainties for emission estimates, it is important to provide estimates of real-life uncertainties in state-of-the-art regional weather analysis systems, and to compare these uncertainties with the OSSE setup. This would be a valuable addition to the conclusions section.

It seems the authors make very optimistic assumptions on the uncertainty of the TEMPO retrievals. Is this justified and are the conclusions sensitive to the choice made for the observation error?

Detailed:

Abstract:

Last line p2: 3.3 to 5 molecules/cmˆ2 ?

p3, l27: "idealized profile setting provided by WRF-Chem." This is unclear and needs to be explained.

p4, top: Please discuss the state augmentation approach for NOx in more detail: how are the emissions perturbed as compared to the inventory, and how is the ensemble constructed.

p5, l1: "We chose the 10 km distance based on sensitivity experiments" How are these experiments done?

p5, l26: "We calculate a layer dependent Box-Air Mass Factor (BAMF) . . . follow the latest version of the NASA . . ." It is unclear from the text how this is done. If the authors compute this themselves, then which RTM is used? How is the geometry computed? What about the terrain parameters. Please provide these details.

p6, l3: "The other parameters are sampled from the model run". Which parameters ?

p6, l14: The mean uncertainty of 7.5 % seems very optimistic. Is this justified somehow? Would the results be very sensitive to this choice?

p6, l29: "scaled to be 70%" A uniform scaling is very idealized, and may be more easily recovered in a DA system than spatially-varying emission perturbations. It seems logical to add also a more random perturbation in emissions to reflect the uncertainty in the emission spatial distribution. Would this impact the conclusions?

p7, l29: Please discuss explicitly the formula to compute the uncertainty. In line 31 the RMS is defined, but "uncertainty" is unclear.

p8, l11: "initial uncertainty of 41.70 mol/(km2Åůhr)". What is this absolute number? Is it an average over the red domain in Fig. 1?

Fig.3: It would be helpful to indicate with symbols at what times an analysis is produced (for the green and blue curves).

p8, l18: Please explain how this correlation is computed.

p9, l3: "future TEMPO NO2 observations will enable us to constrain surface emissions on a city scale". I would claim that this is not fully demonstrated by the authors, because the OSSE setup is still idealized. In particular, the same model is used to construct reality (the nature run), and emissions are idealized, which will lead to too optimistic results. Please formulate the conclusions in a more careful way.

Please also report on typical wind uncertainties which are within reach in present-day high-resolution regional meteorological analysis systems. This may be compared with RMS and uncertainty values as reported in table 2. Is a performance like reported for e.g. ENS-H feasible in reality?

---

## Referee Comment (RC2) · Anonymous Referee #2 · 17 Oct 2016

General comments:

The focus of this paper is to test the ability of estimating emissions of NO2 with data assimilation and future high resolution satellite instruments by doing an Observing System Simulation Experiment (OSSE), since the temporal and spatial variability of NO2 are high and hourly observations at a resolution of at least 5km would be required. I think the title should include "using OSSEs", otherwise it suggests that real data experiments have been carried out.

The abstract and the introductory section are excellent. One of the main experimental conclusions is that the assimilation of meteorology and of NO2 should be done simultaneously, suppporting Kang et al. (2012) who pointed out the need to do simultaneous assimilation of the meteorology and atmospheric CO2 and thus succeeded

(in an OSSE framework) in estimating accurately the surface carbon fluxes even in the absence of any a priori information.

Sections 2-4 are not written clearly enough, in my opinion, at least not for a person who is not very familiar with the details of the subject, like this reviewer. In addition, much of the results revolve around an unrealistic experiment: having the TEMPO NO2 observations at very high spatial and temporal resolution, and at the same time assimilating the winds every 12hr or even 24hr, which, not surprisingly, ruins the results. I would suggest to replace this with a more realistic experiment: Assume that you have access to a meteorology data assimilation performed every hour, so that you have access to the mean winds but not to the uncertainties. Are the wind uncertainties essential? Experiments that we have performed suggest that the wind uncertainties are not so important for CO2, but they may be for NO2 because of the smaller scales involved.

My recommendation is that this paper is important and could be accepted after major revision. Detailed comments:

P3, line 28: "After a spin-up time of 40 hours on the outer domain, the inner domain simulation is initialized and constrained through one-way nesting in both meteorology and chemistry". It is not clear whether this is in addition of the data assimilation, and for how long it is done.

P4 line 1: "We use the RADM2 gas phase chemical mechanism for its simplicity (Stockwell et al., 1990)" I don't know what this means.

Line 12: Since Kang et al. (2011, 2012) were the first to estimate surface fluxes of carbon as evolving parameters, perhaps they should be referenced.

Line 19-24. This is an interesting persistence approach to estimate the emission forecast model. I am not sure whether it will always work in the presence of error spin down or if the surface fluxes vary substantially. It would be good to show how well it works in this case, perhaps under varying surface fluxes.

P5 line 6: "showed" not "suggest"

Lines 13-14: Are these simulated observations obtained from NARR?

Line 24: Does you model changes in the "truth" in the simulated TEMPO observations or are the fluxes fixed?

P6 line 10: It would be good to show a companion figure showing the diurnal cycle.

Line 14 "a mean uncertainty of 7%, which is lower than the 35%..." unclear.

Line 19: AMF: I couldn't find its definition. In general there is a profusion of use of abbreviations whose original definition is difficult to find in the paper. I can deal with Control run (CR) and Nature Run (NR) but then new capitals appear (AR) defined somewhere else, and I get confused. If all the acronyms were defined in the same place it would be OK, but my pdf reader, when I search for the definition of AR, gives me hundreds of words that contain "ar".

P7 line 24: "the 17:00LT assimilation cycle each day" suggests that the assimilation may have been done for many days, and I really don't know yet whether this is true, or just one day was simulated (with constant true emission?).

Along these lines, it would be really strange for a real life researcher to use detailed TEMPO observations every hour and then do atmospheric assimilation only every 12 or even 24 hours. The experiments ENS-O and ENS-T are not realistic (and have horrible acronyms). I would drop these experiments. The results shown in Figure 3 are very obvious: when you assimilate wind observations the uncertainty in the winds becomes much smaller, and the transport errors likewise.

P8 line 7 The acronym of "No meteorological observations assimilated" should NOT be called "BIAS-MET", the errors are much more than a bias!

Fig 3: Label "Local time (hours)"

Fig 4: Labels: "Longitude" "Latitude" in the figure, not in the figure caption.

Fig 5: It confirms that not assimilating the atmospheric winds results in huge errors, as expected.

Fig 6: BIAS-MET is yellow?

[Figure]

---

## Author Comment (AC1) · 17 Feb 2017

We appreciate the referee's comments, which have improved our manuscript, as detailed below:

*General comments: The focus of this paper is to test the ability of estimating emissions of NO2 with data assimilation and future high resolution satellite instruments by doing an Observing System Simulation Experiment (OSSE), since the temporal and spatial variability of NO2 are high and hourly observations at a resolution of at least 5km would be required. I think the title should include "using OSSEs", otherwise it suggests that real data experiments have been carried out.*

We updated the title as "Assimilation of satellite $NO_2$ observations at high spatial resolution using OSSEs"

*The abstract and the introductory section are excellent. One of the main experimental conclusions is that the assimilation of meteorology and of NO2 should be done simultaneously, supporting Kang et al. (2012) who pointed out the need to do simultaneous assimilation of the meteorology and atmospheric CO2 and thus succeeded (in an OSSE framework) in estimating accurately the surface carbon fluxes even in the absence of any a priori information.*

*Sections 2-4 are not written clearly enough, in my opinion, at least not for a person who is not very familiar with the details of the subject, like this reviewer. In addition, much of the results revolve around an unrealistic experiment: having the TEMPO NO2 observations at very high spatial and temporal resolution, and at the same time assimilating the winds every 12hr or even 24hr, which, not surprisingly, ruins the results. I would suggest to replace this with a more realistic experiment: Assume that you have access to a meteorology data assimilation performed every hour, so that you have access to the mean winds but not to the uncertainties. Are the wind uncertainties essential? Experiments that we have performed suggest that the wind uncertainties are not so important for CO2, but they may be for NO2 because of the smaller scales involved.*

We have made major revisions in section 2~4. In section 2, we provide detailed description on the assimilation system including the forecast model, DART facility (with localization settings), initial and boundary condition ensemble, emission update scheme and synthetic meteorological and chemical observations. In section 3, we redesign the assimilation experiment. First, we incorporated MADIS weather observations to increase the volume of wind observations. This enables us to preform hourly joint assimilation of meteorology and chemistry in the run named ENS.1. Following the reviewer's suggestion, we have REA assimilation run in which the meteorology is extracted from an existing reanalysis. REA reinitializes the meteorological state every hour with the best estimate of meteorological states generated by ENS.1. The result discussion and conclusion are also updated accordingly in section 4.

*My recommendation is that this paper is important and could be accepted after major revision. Detailed comments:*

*P3, line 28: "After a spin-up time of 40 hours on the outer domain, the inner domain simulation is initialized and constrained through one-way nesting in both meteorology and chemistry". It is not clear whether this is in addition of the data assimilation, and for how long it is done.*

This part only introduces the domain setup and how simulation on the outer domain serves for the inner domain. There is no assimilation on the outer domain. We describe the experiment details in section 3. The inner domain simulation is initialized on 06:00 am on 2014/07/02 and assimilation starts on 10:00 am 2014/07/02 (P8 line5). We extended the model spin-up time and modified the text as below:

"We use the global chemical model output from MOZART to initialize the chemical simulation on the outer domain and to provide the chemical boundary condition. After a spin-up time of four days on the outer domain, the inner domain simulation is initialized and constrained through one-way nesting in both meteorology and chemistry."

*P4 line 1: "We use the RADM2 gas phase chemical mechanism for its simplicity (Stockwell et al., 1990)" I don't know what this means.*

We provide descriptions of RADM2 chemical mechanism as below.

"We use the widely-used regional acid deposition model version 2 (RADM2) as the gas phase chemical mechanism. There are 59 species and 157 reactions to represent both inorganic and organic chemical reactions under tropospheric conditions. It includes the chemical losses of NOx through reaction with OH radical to form nitric acid, and other $NO_x$ sinks as peroxyacyl nitrates and alkyl nitrate. RADM2 predicts peak NO 2 concentrations and the timing that the peak occurs very well when the mechanism is tested against the representative environmental chamber experiments."

*Line 12: Since Kang et al. (2011, 2012) were the first to estimate surface fluxes of carbon as evolving parameters, perhaps they should be referenced.*

We modified the text as follows in section 2.4:

"By including emissions in the ensemble state vector, emissions are estimated as hourly evolving parameters. Estimation of time-evolving emissions using data assimilation is first presented in carbon fluxes estimation (Kang et al., 2011, 2012). Such an approach provides emission information more than a monthly mean, or an average for a specific time period. For NOx emission estimation problem, emissions in cities show significant variation from urban to suburban. The observed columns show strong spatial variation dominated by emission hotspot as a result of the short lifetime. Combined with high-resolution observations, the goal in this work is to constrain a time-static emission error as we perturb emissions of each hour in the same way."

*Line 19-24. This is an interesting persistence approach to estimate the emission forecast model. I am not sure whether it will always work in the presence of error spin down or if the surface fluxes vary substantially. It would be good to show how well it works in this case, perhaps under varying surface fluxes.*
In all model runs, the surface fluxes are varying from hour to hour. In the OSSE, the emission bias in the control run is -30% uniformly though 24 hours. We agree that in reality, the errors in emissions could vary from hour to hour. So there may not be such a constant relative bias. However, the simplicity of this assumption makes it useful for this initial trial. We hope that other experiments with alternate assumptions will emerge in future research.

*P5 line 6: "showed" not "suggest"*

We changed to "showed".

*Lines 13-14: Are these simulated observations obtained from NARR?*

The simulated meteorological observations are sampled from the nature run (NR, see in section 3). We described the simulated MADIS observations in detail in section 2.5.

*Line 24: Does you model changes in the "truth" in the simulated TEMPO observations or are the fluxes fixed?*

The NO$_x$ emissions are specified by NEI 2011 dataset. This emission provides hourly-varying emission for a typical weekday in summertime. The fluxes does not vary from day to day. NEI 2011 is used as the emission input for nature run without any perturbation. We added this information in section 2.1:

"Anthropogenic emissions for WRF-Chem are from the National Emission Inventory (NEI) 2011 for a typical July weekday 30 at native 4×4 km resolution. The NEI 2011 provides hourly-varying emission for a typical weekday in summertime. The emissions do not vary from day to day."

In section 3, we added as:

"The original NEI 2011 is used as the emission input for the NR without any emission perturbation. We begin by creating a NR and a CR simulation on the outer domain of 12 km resolution (d01) without assimilating observations using a simulation setup as described above in section 2.1. We impose a difference to the CR by using emissions in the CR that are scaled to be 70% of the NR emissions."

*P6 line 10: It would be good to show a companion figure showing the diurnal cycle.*

The diurnal cycle of TEMPO NO$_2$ column is now shown in Figure 4.

*Line 14 "a mean uncertainty of 7%, which is lower than the 35%..." unclear.*

This sentence is not in the revised paper.

*Line 19: AMF: I couldn't find its definition. In general there is a profusion of use of abbreviations whose original definition is difficult to find in the paper. I can deal with Control run (CR) and Nature Run (NR) but then new capitals appear (AR) defined somewhere else, and I get confused. If all the acronyms were defined in the same place it would be OK, but my pdf reader, when I search for the definition of AR, gives me hundreds of words that contain "ar".*

We defined box- air mass factor as BAMF previously, but not AMF. We thank the reviewer for pointing out this issue. We provide the definition of abbreviations in the text as follows.

"Finally, as TEMPO is expected to be operational no sooner than 2018, it is reasonable to expect the retrieval error dominated by air mass factor (AMF) in polluted region will be reduced as a result of future improvements in AMF simulation"

We removed AR which confuses the reader. We define nature run (NR) and control (CR) as below:

"The original NEI 2011 is used as the emission input for the NR without any emission perturbation. We consider the NR as the true atmosphere and sample meteorological and NO2 observations from the NR. The control run (CR) is a parallel model calculation to the NR and suffers from imperfect model input and parametrization. The differences between the NR and the CR in this study are the emission inputs and the initial conditions for the meteorology."

*P7 line 24: "the 17:00LT assimilation cycle each day" suggests that the assimilation may have been done for many days, and I really don't know yet whether this is true, or just one day was simulated (with constant true emission?).*

In the updated experiments, we extended the assimilation to three days longer. We modified the text as below:

"We run assimilation experiments from 10:00 LT 2014/07/02 to 18:00 LT 2014/07/05 with an assimilation window of one hour."

*Along these lines, it would be really strange for a real life researcher to use detailed TEMPO observations every hour and then do atmospheric assimilation only every 12 or even 24 hours. The experiments ENS-O and ENS-T are not realistic (and have horrible acronyms). I would drop these experiments. The results shown in Figure 3 are very obvious: when you assimilate wind observations the uncertainty in the winds becomes much smaller, and the transport errors likewise.*

As indicated above, we redesigned the assimilation experiment as described in section 3. First, we incorporated MADIS weather observations to greatly increase the volume of wind observations. This enables us to preform hourly joint assimilation of meteorology and chemistry in the run named ENS.1. Also we add ENS.2 to determine whether or not concentrations should be updated when observations are assimilated to constrain unobserved emissions. In ENS.3, we use the meteorology of the next day to initialize the CR ensemble so that there is some difference between the CR ensemble mean and the NR in the meteorology. REA reinitializes the meteorological state every hour with the best estimate of meteorological states generated by ENS.1. We hoep the reviewer finds the strategy and acronyms an improvement.

*P8 line 7 The acronym of "No meteorological observations assimilated" should NOT be called "BIAS-MET", the errors are much more than a bias!*

We removed this run with no meteorological observations assimilated in the new experiment design.

*Fig 3: Label "Local time (hours)"*

We added "local time (hours)" in Figure 3~6.

*Fig 4: Labels: "Longitude" "Latitude" in the figure, not in the figure caption.*

The original Figure 4 is no longer in our updated paper.

---

## Author Comment (AC2) · 17 Feb 2017

We thank the reviewer for the time and review. Your feedback has improved our manuscript.

*In their paper, X. Liu and co-authors discuss the impact of future geostationary NO2 observations on air quality models and the potential of these observations to improve estimates of emissions.*

*General assessment: The highlight of this paper is a documentation of the sensitivity of NOx emission estimates on the quality of the meteorological fields and the winds in particular. As shown, these issues are especially critical when the resolution of the observations and the models advances to the km scale. This sensitivity is demonstrated in a clear way, and to my opinion is an important aspect to keep in mind for the development of future regional analysis systems. As such, I am in favor of publishing these results in ACP.*

*The paper is well written, has a good introduction, but the overview of the method C1 and tools is very short and condensed. The implementation of the OSSE setup is not provided with enough detail for other groups to repeat the experiment, see my detailed comments below. I therefore would urge the authors to expand these parts of the paper.*

We have made major revisions in section 2~4. In section 2, we provide detailed description on the assimilation system including the forecast model, DART facility (with localization settings), initial and boundary condition ensemble, emission update scheme and synthetic meteorological and chemical observations. In section 3, we redesign the assimilation experiment.

*In order to judge the importance of the meteorological uncertainties for emission estimates, it is important to provide estimates of real-life uncertainties in state-of-the-art regional weather analysis systems, and to compare these uncertainties with the OSSE setup. This would be a valuable addition to the conclusions section.*

We agree that it's important to compare the identified wind uncertainties in this work with state-of-the-art regional weather analysis. The available atmospheric reanalysis products (https://climatedataguide.ucar.edu/climate-data/atmospheric-reanalysis-overview-comparison-tables) provide a dynamically consistent estimate of the climate state at each time step. These assimilation scheme and models ingest all available observations every 6-12 hours and produce global and regional reanalysis at mesoscales (above 32 km). The products with such assimilation frequency and spatial resolution do not provide analysis uncertainties at the fine scale as in this work, which is hourly assimilation at the resolution of 3 km. In this work, we downscale NARR (32 km) to initialize the WRF forecast and use WRFDA to generate the ensemble. WRFDA system uses a state of the art method for generating the ensembles/quantifying the meteorological uncertainty. It uses the NMC method to generate the static error covariance and then uses the static error covariance to perturb the meteorological fields. Since the WRF meteorological forecast errors are comparable to other regional system, we would expect the uncertainties of the associated ensembles to be representative and reasonable. So we revised the text as follows:

"For meteorology ensemble, random perturbations were added to each member by sampling the NCEP background error covariance using WRF Data Assimilation System (http://www2.mmm.ucar.edu/wrf/users/wrfda)."

Since we redesign our experiment to assimilate meteorological observations hourly, the meteorological uncertainties becomes small enough to allow chemical observations to constrain emissions successfully in our results. Then we analyze whether the wind uncertainties are important for emission estimation in REA. Based on our result in REA run, we conclude as following:

"Last but not least our results show that the information of wind *uncertainties* is not important for $NO_x$ emission estimation. Assimilations of $NO_2$ only with the meteorology constrained from an hourly weather assimilation product would perform as well as the joint assimilation of meteorology and chemistry."

*It seems the authors make very optimistic assumptions on the uncertainty of the TEMPO retrievals. Is this justified and are the conclusions sensitive to the choice made for the observation error?*

In ensemble assimilation method, the adjustment on the emissions depends on the relative magnitude of TEMPO observation uncertainties and the prior emission uncertainties. We agree that the uncertainty of TEMPO retrieval in this study is optimistic compared to current OMI retrieval. However, previously we have conducted experiments with TEMPO observation of 30%. We find the emission estimation results not promising because the observation uncertainties are too large to let the observations influence the emissions. In addition, the prior emission uncertainty is set to be 30% in our OSSE. This can also be very optimistic given the dispute on the accuracy of bottom-up inventory. In real application, with a reasonable estimation of prior emission uncertainty which will be higher than 30%, it is highly possible that more weights will be given to TEMPO observations to adjust the emissions. The choice of TEMPO retrieval uncertainty in this study is limited by the estimation of the prior emission uncertainties. In real application, even if the reported TEMPO uncertainty is too large to adjust the emissions, the conclusions of our OSSEs on the optimal approach of $NO_x$ emission estimation will not change.

*Last line p2: 3.3 to 5 molecules/cm^2 ?*

We change the text as, "biases of 34% (3.3 to 5.0 $\times 10^{15}$ molecules/cm$^2$) are found in the modeled average $NO_2$ column over Los Angeles at resolutions of 96 km compared with 12 km."

*p3, l27: "idealized profile setting provided by WRF-Chem." This is unclear and needs to be explained.*

In our updated experiments we initialize WRF-Chem using the global chemical transport model outputs that will provide a realistic initial condition for chemical species. We modified the text as, "We use the global chemical model output from MOZART to initialize the chemical simulation on the outer domain and to provide the chemical boundary condition."

*p4, top: Please discuss the state augmentation approach for NOx in more detail: how are the emissions perturbed as compared to the inventory, and how is the ensemble constructed.*

Compared with the NEI 2011 which is referred as the truth, the CR emissions are perturbed by adding a 30% low bias uniformly in temporal and spatial scale. This is used as the ensemble mean of emissions in the assimilation process to construct the emission ensemble. Each ensemble member is then generated by adding a perturbation to the ensemble mean. The resulting emission ensemble in this work shows ~35% uncertainty presented by the ensemble standard devition. The correlation between emission perturbations of two grid point is modeled by a simple isotropic exponential decay function with a characteristic correlation length of 50 km.

We discuss the state augmentation approach for emissions in section 2.4 in detail. We described the ensemble construction in section 2.3. To better explain the emission perturbation, we added the text as below:

"We consider the NR as the true atmosphere and sample meteorological and $NO_2$ observations from the NR. The control run (CR) is a parallel model calculation to the NR and suffers from imperfect model input and parametrization. The differences between the NR and the CR in this study are the emission inputs and the initial conditions for the meteorology. … . In all experimental runs, we bias the CR initial emissions to be 30% below the reference model and examine the ability of the assimilation to recover the reference emissions."

*p5, l1: "We chose the 10 km distance based on sensitivity experiments" How are these experiments done?*

We changed the text to the following: We perform univariate experiment by tuning the localization distance from 5km, 10km, 20km to 50km and keeping other settings in the system unchanged. By comparing the posterior emission RMSE of REF (described in section 3) of all experiments, we find the smallest RMSE with the localization distance of 10km.

*p5, l26: "We calculate a layer dependent Box-Air Mass Factor (BAMF) : : : follow the latest version of the NASA : : :" It is unclear from the text how this is done. If the authors compute this themselves, then which RTM is used? How is the geometry computed?*
*What about the terrain parameters. Please provide these details.*

We didn't calculate BAMF by running a RTM directly as it is not required when we follow the method in (Bucsela et al., 2013). "The scattering weights are computed and stored a priori in six-dimensional look-up tables (LUT) generated from a RTM. The six LUT parameters are solar zenith angle (SZA), viewing zenith angle (VZA), relative azimuth angle (RAA), terrain reflectivity (Rt), terrain pressure (Pt), and atmospheric pressure level, (p). ". We modified the text as below:

"The geometry related parameters (SZA, VZA and RAA) are computed hourly for each TEMPO observation using Matlab functions sun_position.m and geodetic2aer.m with inputs of the location and time of each TEMPO observation, and the location (36.5°N, 100°W) and altitude (35,786 km) of the TEMPO sensor. The terrain reflectivity and terrain pressure are sampled from the WRF-Chem nature run (NR, see section 3) for each TEMPO pixel."

*p6, l3: "The other parameters are sampled from the model run". Which parameters?*

Other parameters indicate terrain reflectivity and terrain pressure. We clarified this in the updated text.

*p6, l14: The mean uncertainty of 7.5 % seems very optimistic. Is this justified somehow?*
*Would the results be very sensitive to this choice?*

We agree that it is very optimistic to set the TEMPO observation uncertainty as 7.5%, which is the lower bound of current observation error for OMI $NO_2$ tropospheric column product. This paper focuses on the optimal approach for $NO_x$ emission estimations from TEMPO. It is difficult to separate systematic and random contributions to the uncertainty in TEMPO measurements. Future experiments should explore variations in systematic and random uncertainties in the measurements.

*p6, l29: "scaled to be 70%" A uniform scaling is very idealized, and may be more easily recovered in a DA system than spatially-varying emission perturbations. It seems logical to add also a more random*

*perturbation in emissions to reflect the uncertainty in the emission spatial distribution. Would this impact the conclusions?*

The conclusions in this work will not change by using a spatial-varying emission perturbation. We agree that a uniform emission perturbation is idealized. Success in recovering emissions with a uniform scaling can't not guarantee the success to recover emissions with spatially-varying perturbations. However, the approach that didn't work in the ideal case will fail in the realistic scenario with spatially-inhomogeneous emission perturbations. In our new experiments, we show the shortcomings of some emission estimation strategy in the scenario of ideal emission perturbation. We added the following text in section 5.

"In our OSSE, we start with an ideal case of a priori emissions, which is spatially-uniform 70% of true emissions. Success in recovering emissions with a uniform scaling can't not guarantee the success to recover emissions with spatially-varying perturbations. However, the approach that didn't work in the ideal case will fail in the realistic scenario with spatially-inhomogeneous emission perturbations. The degradation of emission estimation in the ENS.2 and ENS.3 will still hold for correcting emission errors under more complex circumstances, such as spatially- or temporally- varying emission errors. For future work, further analysis is required to examine whether spatially-varying emission errors can be reduced under the condition of low wind RMSE."

*p7, l29: Please discuss explicitly the formula to compute the uncertainty. In line 31 the RMS is defined, but "uncertainty" is unclear.*

We added the text as follows:
"We also analyze the uncertainty of the prior and posterior estimates. The uncertainty is expressed by the 1-σ standard deviation of the ensemble."

*p8, l11: "initial uncertainty of 41.70 mol/(km$^2$hr)". What is this absolute number? Is it an average over the red domain in Fig. 1?*

The sentence is removed in our updated discussion section 4.

*Fig.3: It would be helpful to indicate with symbols at what times an analysis is produced (for the green and blue curves).*

Since we updated the experiments in this paper, the original Figure 3 is removed.

*p8, l18: Please explain how this correlation is computed.*

We removed the discussion of correlation in our updated paper.
*P10, l3: "future TEMPO NO2 observations will enable us to constrain surface emissions on a city scale". I would claim that this is not fully demonstrated by the authors, because the OSSE setup is still idealized. In particular, the same model is used to construct reality (the nature run), and emissions are idealized, which will lead to too optimistic results. Please formulate the conclusions in a more careful way.*
*Please also report on typical wind uncertainties which are within reach in present-day high-resolution regional meteorological analysis systems. This may be compared with RMS and uncertainty values as reported in table 2. Is a performance like reported for e.g. ENS-H feasible in reality?*

We clarified our point as follows:

"We test the emission constraints from TEMPO $NO_2$ observations in an ideal case assuming no errors associated with the modeled meteorology. In the experiment of joint assimilation of meteorology and chemical $NO_2$, we find that the emission estimation is successful in the morning but degrades in the afternoon when the prior wind RMSE grows above 1 m/s. Considering the dependence of errors in estimated emissions on the wind forecast errors, we recommend to guarantee the accuracy in modeled wind with RMSE below 1 m/s for the success of chemical assimilation to infer emissions at the scale of our model grid."

Reference:

Dee, Dick, Fasullo, John, Shea, Dennis, Walsh, John & National Center for Atmospheric Research Staff (Eds). Last modified 21 Jan 2016. "The Climate Data Guide: Atmospheric Reanalysis: Overview & Comparison Tables." Retrieved from https://climatedataguide.ucar.edu/climate-data/atmospheric-reanalysis-overview-comparison-tables.

---

## Referee Report (RR1)

General comments:

The highlight of this paper is a documentation of the sensitivity of NOx emission estimates on the quality of the meteorological fields and the winds in particular. As shown, these issues are especially critical when the resolution of the observations and the models advances to the km scale. This sensitivity is demonstrated in a clear way, and to my opinion is an important aspect to keep in mind for the development of future regional analysis systems. As such, I am still in favour of publishing these results in ACP.

The revised document has a completely new set of figures and new experiments, which provide more insight into the behaviour of the analysis results. This has improved the paper considerably compared to the ACPD version. Despite this, there are still aspects of the paper that can be improved, as documented below.

It would be good to check the English spelling throughout the document, especially for the parts which changed since the previous version of the document.

Generally the abstract is not well written and not well structured and should be revised. It should clearly state the aim of the study, the tools and simulations, and the conclusions. See my comments below for further details.

Specific comments:

- Abstract: The first 10 lines of the abstract would be better placed in the introduction, and may be largely removed.

- Abstract: "Accurate observation and modelling of these variations require spatial resolution of order 4 km." This fits well with TEMPO, but for me it is unclear what the quoted length scale of 4km is based on. It is not a result of the paper so I suggest to remove it from the abstract.

- Abstract: The aim of the study can be provided in more detail. In particular the investigation of the sensitivity of emission inversions on the accuracy of the wind analyses.

- Abstract: What is missing is a description of what is done (the OSSE experiments and tools). The ingredients of the OSSE setup should be explained in a few sentences.

- Abstract: "Last but not least our results show that the information of wind uncertainties is not important for NOx emission estimation." This is unclear. First it is claimed that it is essential that the wind RMSE should be below 1 m/s, which seems to contradict the statement that the wind uncertainty is not important !? See also below.

- Introduction: Is well written, clearly states the aim of the study and contains a good set of references. No further comments.

- Sec 2.1, p4, l12: "RADM2 predicts peak NO2 concentrations and the timing that the peak occurs very well when the mechanism is tested against the representative environmental chamber experiments". Is there a reference ?

- p5, l13: "In our next paper we will discuss the potential benefit from assimilations of chemical observations to improve the meteorological fields." I would suggest to remove this line: it is not interesting for the reader to know about future studies.

- Tables 2 and 3: For people not familiar with DART and WRFDA these tables are difficult to understand, e.g. terms like "cv_options" and "je_factor" is not meaningful. It would be good to explain these options in the text.

- Section 2.4: The description of the emission updates is described in a confusing way. First, time-evolving emissions are introduced. Then it is mentioned that "time-static" emissions are the aim of the study. After that, emission scaling factors are introduced, which change after each assimilation, which seems to contradict the "time-static" remark. Please re-write and explain more clearly how emissions are analysed.

- p7, l1: "We calculate a layer dependent Box-Air Mass Factor" This statement is a bit confusing as it suggests that a radiative transfer code is run. I understand that lookup tables from Bucsela et al are used. Please formulate more clearly.

- p7, l6: "We assume clear-sky conditions for all observing scenes." Why is this done? This will increase the number of observations artificially.

- Table 4: The normal distribution N( 200%, 100%). What does this mean? Why are there two percentages? This seems to contradict the notation of normal distributions as given in p7, l32, where I would expect the first number to be the nature run value.

- p10, l9: "succeeds in recovering the true emissions (Figure 5)." I guess you mean Figure 6? The order of the figures should follow the order of the references to the figures in the text (5 <-> 6).

- p11, l5-10: The emission update scheme is explained here. The "scaling factor for t-1" seems to be inconsistent with the discussion in section 2.4, which only mentions updates for e_{i+1}. The two texts seem to be inconsistent. Please explain the emission update process more clearly.

- p13, l24: "Last but not least our results show that the information of wind uncertainties is not important for NOx emission estimation." I guess that the combined wind/NO2 analysis could also lead to adjustments of the winds which give a better match between observed NO2 concentrations and the location of the emission sources. The ensemble, with a spread equal to the wind uncertainty, would allow for such adjustments and could in principle outperform the system where the winds are prescribed. It seem the authors claim that such benefits are not significant?

- p13, l26: "Assimilations of NO2 only with the meteorology constrained from an hourly weather assimilation product would perform as well as the joint assimilation of meteorology and chemistry."
Do you suggest that a CTM with prescribed meteorological analyses will perform as well as a fully integrated chemistry-meteorology analysis system?

- p13, l34: "will fail" -> will also fail

---

## Author Response (AR2)

We thank the reviewer for the comments which greatly improves our manuscript.

*General comments:*

*The highlight of this paper is a documentation of the sensitivity of NOx emission estimates on the quality of the meteorological fields and the winds in particular. As shown, these issues are especially critical when the resolution of the observations and the models advances to the km scale. This sensitivity is demonstrated in a clear way, and to my opinion is an important aspect to keep in mind for the development of future regional analysis systems. As such, I am still in favour of publishing these results in ACP.*

*The revised document has a completely new set of figures and new experiments, which provide more insight into the behaviour of the analysis results. This has improved the paper considerably compared to the ACPD version. Despite this, there are still aspects of the paper that can be improved, as documented below.*

*It would be good to check the English spelling throughout the document, especially for the parts which changed since the previous version of the document.*

We checked the English spelling and grammar throughout the manuscript.

*Generally the abstract is not well written and not well structured and should be revised. It should clearly state the aim of the study, the tools and simulations, and the conclusions. See my comments below for further details.*

*Specific comments:*

*- Abstract: The first 10 lines of the abstract would be better placed in the introduction, and may be largely removed.*

*- Abstract: "Accurate observation and modelling of these variations require spatial resolution of order 4 km." This fits well with TEMPO, but for me it is unclear what the quoted length scale of 4km is based on. It is not a result of the paper so I suggest to remove it from the abstract.*

We removed this statement from the abstract.

*- Abstract: The aim of the study can be provided in more detail. In particular the investigation of the sensitivity of emission inversions on the accuracy of the wind analyses.*

*- Abstract: What is missing is a description of what is done (the OSSE experiments and tools). The ingredients of the OSSE setup should be explained in a few sentences.*

*- Abstract: "Last but not least our results show that the information of wind uncertainties is not important for NOx emission estimation." This is unclear. First it is claimed that it is essential that the wind RMSE should be below 1 m/s, which seems to contradict the statement that the wind uncertainty is not important !? See also below.*

We have revised the abstract to address these comments and state our objectives and conclusions more clearly.

*- Introduction: Is well written, clearly states the aim of the study and contains a good set of references. No further comments.*

*- Sec 2.1, p4, l12: "RADM2 predicts peak NO2 concentrations and the timing that the peak occurs very well when the mechanism is tested against the representative environmental chamber experiments". Is there a reference?*

The statement is cited from the reference (Stockwell et al., 1990). In the revision, we decided to remove this sentence.

*- p5, l13: "In our next paper we will discuss the potential benefit from assimilations of chemical observations to improve the meteorological fields." I would suggest to remove this line: it is not interesting for the reader to know about future studies.*

We removed this line from the manuscript.

*- Tables 2 and 3: For people not familiar with DART and WRFDA these tables are difficult to understand, e.g. terms like "cv_options" and "je_factor" is not meaningful. It would be good to explain these options in the text.*

We explained the "cv_options" and "je_factor" in the text as below:

"The options in WRFDA settings is summarized in Table 3. The parameter cv_option indicates the background error options in WRFDA. With cv_option = 3, we use the NCEP background error covariance, which is estimated in grid space by what has become known as the NMC method. The statistics are estimated with the differences of 24 and 48-hour GFS forecasts with T170 resolution, valid at the same time for 357 cases, distributed over a period of one year. The parameter je_factor is the ensemble covariance weighting factor. This factor controls the weighting component of ensemble and static covariances."

*- Section 2.4: The description of the emission updates is described in a confusing way. First, time-evolving emissions are introduced. Then it is mentioned that "time-static" emissions are the aim of the study. After that, emission scaling factors are introduced, which change after each assimilation, which seems to contradict the "time-static" remark. Please re-write and explain more clearly how emissions are analysed.*

We have attempted to clarify the logic.

The $NO_x$ emissions in the inventory are varying hour by hour. With an hourly assimilation frequency, the emission scaling factors are also updated in each hour. In the OSSE setup, the emission error, which is the difference between the truth and the prior emission, is initialized with a constant fraction error. The prior emission is initialized at 70% of the truth at each hour. We modified the text as below:

"The goal of this work is to constrain hourly evolving emissions at the native model resolution. Here we start with a simple case in which the emission error is a constant fraction at all times of day with the prior

emissions set as 70% of the truth and we investigate the ability of assimilation to recover the original emissions."

*- p7, l1: "We calculate a layer dependent Box-Air Mass Factor" This statement is a bit confusing as it suggests that a radiative transfer code is run. I understand that lookup tables from Bucsela et al are used. Please formulate more clearly.*

We modified the text as below:

"In $NO_2$ retrieval algorithm, a layer dependent Box-Air Mass Factor (BAMF) represents the sensitivity of the retrieved $NO_2$ in a specific layer to the true value in the atmosphere."

"Without running a radiative transfer code, the elements of the BAMF vector are computed as a function of solar zenith angle (SZA), viewing zenith angle (VZA), relative azimuth angle (RAA), terrain reflectivity (Rt), terrain pressure (Pt), atmospheric pressure level, (p) and the $NO_2$ profile (Bucsela et al., 2013)."

*- p7, l6: "We assume clear-sky conditions for all observing scenes." Why is this done? This will increase the number of observations artificially.*

We agree this artificially increases the number of observations. However, it provides a useful measure of the best possible performance.

*- Table 4: The normal distribution N( 200%, 100%). What does this mean? Why are there two percentages? This seems to contradict the notation of normal distributions as given in p7, l32, where I would expect the first number to be the nature run value.*

Table 4 shows the relative uncertainty $\sigma_{rel}$ for clean and polluted scenario. We set the relative uncertainty as a random draw from a Gaussian distribution to avoid using a fixed value, which mimics the real case. The notation given in p7, l32 has the absolute observation uncertainty σ instead of the relative uncertainty $\sigma_{rel}$. σ is the observation error standard deviation computed as $\sigma = y^{tr} \cdot \sigma_{rel}$. We added this information in Table 4:

"Table 4. Relative observation uncertainty $\sigma_{rel}$ in synthetic TEMPO $NO_2$ column for each scenario."

*- p10, l9: "succeeds in recovering the true emissions (Figure 5)." I guess you mean Figure 6? The order of the figures should follow the order of the references to the figures in the text (5 <-> 6).*

We updated the order of the figures to follow the order of the references to the figures.

*- p11, l5-10: The emission update scheme is explained here. The "scaling factor for t-1" seems to be inconsistent with the discussion in section 2.4, which only mentions updates for e_{i+1}. The two texts seem to be inconsistent. Please explain the emission update process more clearly.*

We updated the text in section 2.4 as follows:

"A challenge for updating the emissions in the augmented state vector is the absence of an emission forecast model to evolve the emission variables forward in time. The bottom-up inventory to be optimized

provides hourly-resolved emissions for each model grid point. Instead of treating the emission variables of each hour at a specific location as independent parameters, we update the emission scaling factors at each assimilation cycle. In our emission update scheme, the TEMPO $NO_2$ observations at time i are assimilated to generate a scaling factor for emissions at time i-1. In this way, the model-observation difference in the $NO_2$ column will correct the emission of an hour ago instead of the current emission. This approach is reasonable because errors in $NO_2$ concentration result from errors in previous emissions. Considering the short $NO_2$ lifetime of three hours in summer daytime, emissions from the previous hour have a large contribution to the $NO_2$ total mass at the current time. For a given model grid point, we define the emissions of truth ($e_{i-1}^t$), prior ($e_{i-1}^{prior}$) and posterior ($e_{i-1}^{post}$) at time $i-1$. Since we start the assimilation with 70% of true emissions, we have $e_m^{prior} = 0.7e_m^t$ for any time m. After assimilating observations at time i, we compute the scaling factor ($S_{i-1}$) for emissions at time $i-1$ as follows: $S_{i-1} = e_{i-1}^{post}/e_{i-1}^{prior}$. Then we update the prior emissions at time i as $e_i^{prior} = S_{i-1} \cdot e_i^{prior}$. This prescription enables us to derive spatial 2-D emission scaling factors which play the role of an emission forecast model."

*- p13, l24: "Last but not least our results show that the information of wind uncertainties is not important for NOx emission estimation." I guess that the combined wind/NO2 analysis could also lead to adjustments of the winds which give a better match between observed NO2 concentrations and the location of the emission sources. The ensemble, with a spread equal to the wind uncertainty, would allow for such adjustments and could in principle outperform the system where the winds are prescribed. It seem the authors claim that such benefits are not significant?*

In our OSSE system, a large volume of wind observations are assimilated to constrain wind in the boundary layer. Under such condition, the wind uncertainties are constrained under 1 m/s for most of the time and show peaks up to 1.5 m/s. Such uncertainties are too small to affect chemical assimilation performance as we find little difference in the emission estimation between ENS.1 and REA. We agree on the comment that the adjustment of the winds could benefit the comparison of observed $NO_2$ concentrations. However, in this assimilation system the wind uncertainties are constrained by wind observations only. With an ensemble of wind, the covariance between wind and $NO_2$ concentrations can potentially provide useful information to adjust winds. Such covariance is not used in ENS.1 as we only allow wind observations to affect wind analysis. In regards to the adjustment of wind from combined wind/$NO_2$ analysis, we will further discuss this in our next paper. In addition, this issue does not conflict with our current finding on the impact of wind uncertainties on $NO_x$ emission estimation. We modified the text as below:

"We would like to point out that the covariance of error statistics between wind and $NO_2$ are not utilized in the OSSE assimilation in this paper. Results on carbon and weather assimilation show that the variable localization scheme we use zeroes out the background error covariance among prognostic variables that are not physically related, thus reducing sampling errors (Kang et al., 2011). Specifically, they find that covariance between carbon fluxes and meteorological variables should be neglected. However, the same result might not obtain for short-lived chemicals. The extent to which chemical observations can be used to improve the assimilation of meteorological variables and vice-versa in a situation where we do not zero the covariance in the errors should be pursued in future research."

*- p13, l26: "Assimilations of NO₂ only with the meteorology constrained from an hourly weather assimilation product would perform as well as the joint assimilation of meteorology and chemistry."*

*Do you suggest that a CTM with prescribed meteorological analyses will perform as well as a fully integrated chemistry-meteorology analysis system?*

We draw such conclusions based on our OSSE result with specific setup options. One important treatment in our OSSE is that we exclude the interaction between chemistry and meteorology during the filtering step. Under this condition, the ensemble statistics between chemistry and meteorology in the integrated chemistry-meteorology assimilation system is not utilized. Our results show that a CTM with prescribed meteorological analyses performs as well as such an integrated system. We modified the text in the response to the previous comment.

*- p13, l34: "will fail" -> will also fail*

We updated the text as below:

[revised manuscript text omitted]